# Energy-saving and product-oriented hydrogen peroxide electrosynthesis enabled by electrochemistry pairing and product engineering

Jun Qi[1], Yadong Du[1], Qi Yang [1] ✉, Na Jiang[1], Jiachun Li[1], Yi Ma[1], Yangjun Ma[1], Xin Zhao[1] & Jieshan Qiu[1] ✉

Hydrogen peroxide ($H_2O_2$) electrosynthesis through oxygen reduction reaction (ORR) is drawing worldwide attention, whereas suffering seriously from the sluggish oxygen evolution reaction (OER) and the difficult extraction of thermodynamically unstable $H_2O_2$. Herein, we present an electrosynthesis protocol involving coupling ORR-to-$H_2O_2$ with waste polyethylene terephthalate (PET) upcycling and the first $H_2O_2$ conversion strategy. Ni-Mn bimetal- and onion carbon-based catalysts are designed to catalyze ORR-to-$H_2O_2$ and ethylene glycol electrooxidation with the Faradaic efficiency of 97.5% ($H_2O_2$) and 93.0% (formate). This electrolysis system runs successfully at only 0.927 V to achieve an industrial-scale current density of 400 mA cm$^{-2}$, surpassing all reported $H_2O_2$ electrosynthesis systems. $H_2O_2$ product is upgraded through two downstream routes of converting $H_2O_2$ into sodium perborate and dibenzoyl peroxide. Techno-economic evolution highlights the high gross profit of the ORR ‖ PET upcycling protocol over HER ‖ PET upcycling and ORR ‖ OER. This work provides an energy-saving methodology for the electrosynthesis of $H_2O_2$ and other chemicals.

Hydrogen peroxide ($H_2O_2$) as an excellent oxidant is widely used in fields such as papermaking, wastewater treatment, medicine, and textiles[1–4]. The traditional anthraquinone process for $H_2O_2$ synthesis adopted in current industrial production suffers from the complex process, serious pollution, and high energy consumption[5–7]. Electricity-driven oxygen reduction reaction (ORR) to $H_2O_2$ (ORR-to-$H_2O_2$) based on a 2e$^-$ transfer path is widely accepted as a promising direction for $H_2O_2$ electrosynthesis due to multiple merits, e.g., mild operating conditions, low voltage directly provided by the renewable power, rich raw material of oxygen, high product selectivity and yield. This booming field experiences rapid development benefitting from catalyst design and mechanism study. For example, metal-free carbon materials[8] (e.g., boron-doped carbon black[9], carbon nanotubes[10]),

metal single-atom catalysts (e.g., Co–N–C[11,12], Pd–N–C matrix composites[13]), and transition metal hybrids (e.g., NiB$_x$[14], CoSe[15]), were developed and achieved high selectivity and yield for ORR-to-$H_2O_2$. Catalysts with low-cost, uniform loading, and long-term stability still deserve further attention.

However, current research mainly focuses on catalyst synthesis, whereas the extraction of $H_2O_2$ as a product from the electrolyte has never been involved. Product separation that usually accounts for more than 50% of the system cost in industrial production matters fundamentally for the practical use[16,17]. High-efficiency extraction of $H_2O_2$ faces the following challenges: (1) the thermodynamic instability of $H_2O_2$; (2) a more complicated and energy-intensive separation process of $H_2O_2$ in the liquid electrolyte than the gaseous product of

[1]State Key Laboratory of Chemical Resource Engineering, College of Chemical Engineering, Beijing University of Chemical Technology, Beijing 100029, P. R. China. ✉e-mail: qi.yang@mail.buct.edu.cn; qiujs@mail.buct.edu.cn

water splitting. Therefore, it is urgent to develop an effective strategy for $H_2O_2$ product extraction or conversion to extend the limited research chain of current $H_2O_2$ electrosynthesis.

Another challenge faced by $H_2O_2$ electrosynthesis is the high anodic potential of oxygen evolution reaction (OER) and the low-value oxygen. Anodic small molecular oxidation reactions (AORs) have attracted intensive attention as an alternative reaction to OER, due to the lower operating potential (thermodynamically favorable)[18–20]. Moreover, compared with the low-value oxygen, AORs products as fine chemicals possess high value and can significantly enhance the economic competitiveness of the $H_2O_2$ electrosynthesis system[21]. For example, glycerol[22–24], methanol[25–27], glucose[28], and 5-hydroxymethylfurfural[29–31], have been used as reactants for proceeding AORs. Other than these chemicals, waste plastics could also participate in AORs with the bonus of transferring plastics to value-added chemicals. The waste polyethylene terephthalate (PET) can be depolymerized into p-phthalic acid (PTA) and ethylene glycol (EG), which can be used to get the PTA product and proceed the EG oxidation (EOR) to produce formate or potassium diformate (KDF)[32,33]. This helps to reshape the plastic life cycle and alleviate the plastic crisis[34–36]. Up to now, most of the application scenarios of AORs focused on the conjunction with the green $H_2$ production, while rare attention has been paid to the design of the electrosynthesis system coupling with ORR-to-$H_2O_2$[37,38].

To address the energy consumption issue faced by current $H_2O_2$ electrosynthesis, we electrochemically paired it with anodic PET upcycling (Fig. 1a). This electrosynthesis protocol contributes to the apparently reduced energy consumption as reflected by the low cell voltages of 0.712, 0.794, and 0.927 V for achieving the industrial-scale current densities of 100, 200, and 400 mA cm$^{-2}$, outperforming all reported $H_2O_2$ electrosynthesis systems. In situ Raman and X-ray absorption near edge structure spectra (XANES) uncover the mechanism and active center of catalyst. For the first time, the $H_2O_2$ electrosynthesis field was pushed forward from current synthesis step to the downstream product conversion by converting the thermodynamically unstable $H_2O_2$ to dibenzoyl peroxide (BPO) and sodium peroxyborate (SPB) in electrolyte (Fig. 1b). Techno-economic evaluation was conducted to present the energy-saving and profit advantages.

## Results

### Synthesis and characterizations of EOR catalysts

Embedding high-valence metals (Mn, V, W, Mo, Cr)[39,40] is an effective strategy to regulate the adsorption strength of 3$d$ metals (Ni, Co, Fe) on reaction intermediates, thereby enhancing the catalytic activity for OER and AORs[22,37,41]. Transition metal hybrids also show higher activity than transition metal oxides, thanks to their excellent electronic transmission ability. Here, a Ni, Mn bimetallic organic framework supported on the nickel foam (Ni$_1$Mn$_1$-MOF/NF) was designed via a solvothermal method, followed by further selenization to generate the final catalyst (Ni$_1$Mn$_1$-MOF-Se/NF, Fig. 2a, Supplementary Figs. 1 and 2). X-ray diffraction (XRD) pattern detects NiSe and Ni$_{0.85}$Se with no MoSe phase, indicating that the Mn element exists at the form of lattice doping rather than heterostructure (Supplementary Fig. 3). Scanning electron microscopy (SEM) images reveal that the coral-like structure composed of ultra-thin nanosheets, which originates from the reconstruction of Ni$_1$Mn$_1$-MOF-Se/NF with a nano-array structure (Fig. 2b, Supplementary Figs. 4 and 5). Rich channel networks and rough surfaces are beneficial for the mass transfer of electrolyte and the exposure of active sites. CV activation causes the oxidation of Ni and Mn in the Ni$_1$Mn$_1$-MOF-Se/NF catalyst, demonstrating the reconstruction on the catalyst surface. Element mapping images of Ni$_1$Mn$_1$-MOF-Se/NF after electrochemical activation indicate that the Se element of the catalyst surface was replaced by O (Supplementary Fig. 5)[25,26]. XRD patterns (Supplementary Fig. 3) indicate that the Ni$_1$Mn$_1$-MOF-Se/NF catalyst after activation still shows the presence of NiSe and Ni$_{0.85}$Se, indicating that the Se element in the solid phase remains stable. Transmission electron microscopy (TEM) and high-resolution TEM (HRTEM) images (Fig. 2c, d and Supplementary Figs. 6 and 7) reveal the two-dimensional structure featured by apparent lattice fringes with d-spaces of about 0.270 and 0.202 nm, corresponding to the (101) and (102) planes of Ni$_{0.85}$Se. Element mappings show the homogeneous distribution of Ni, Mn, Se, and O (Fig. 2e) in the Ni$_1$Mn$_1$-MOF-Se/NF catalyst, while the Ni$_1$Mn$_1$-MOF-Se/NF catalyst after electrochemical activation underwent partial aggregation of Se and Mn due to the reconstruction featured by the oxidation of Ni and Mn (Fig. 2e and Supplementary Fig. 5). Inductively coupled optical emission spectrometric (ICP-OES) measurements reveal the Ni/Mn quality ratio of 10:1

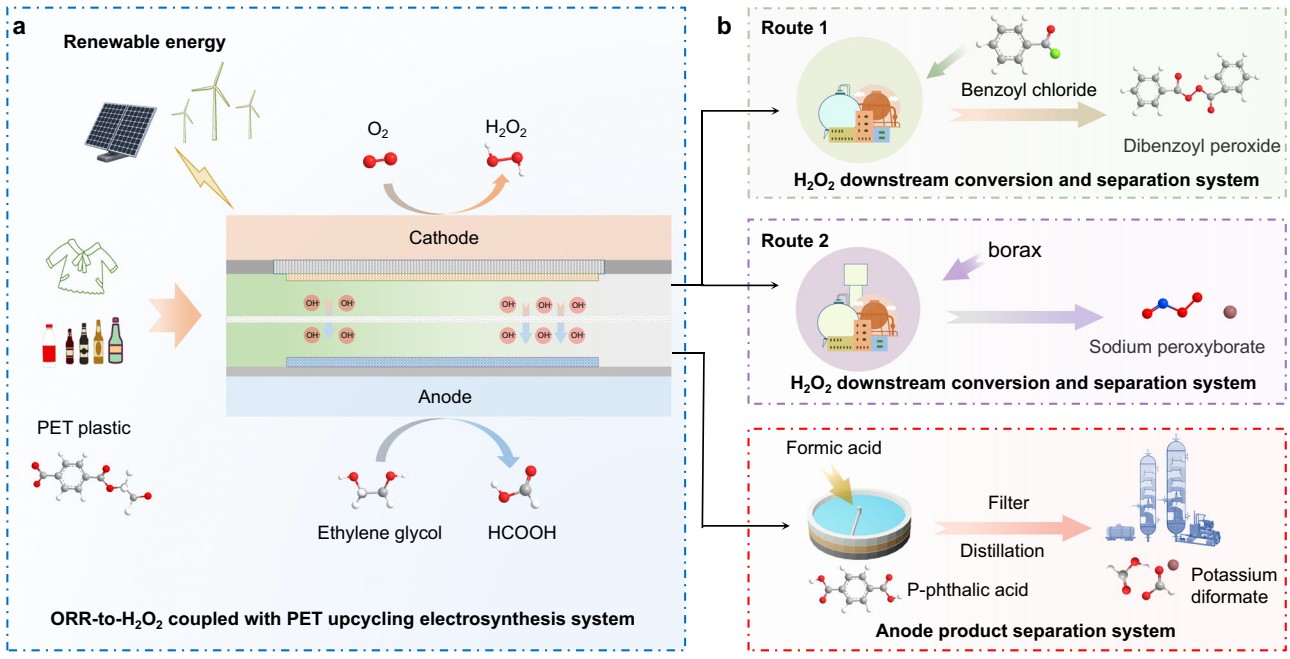

**Fig. 1 | Schematic illustration of the electrolysis system coupling ORR-to-$H_2O_2$ with PET upcycling. a** The electrolysis system of $H_2O_2$ electrosynthesis coupling with PET upcycling. **b** Product separation and conversion systems.

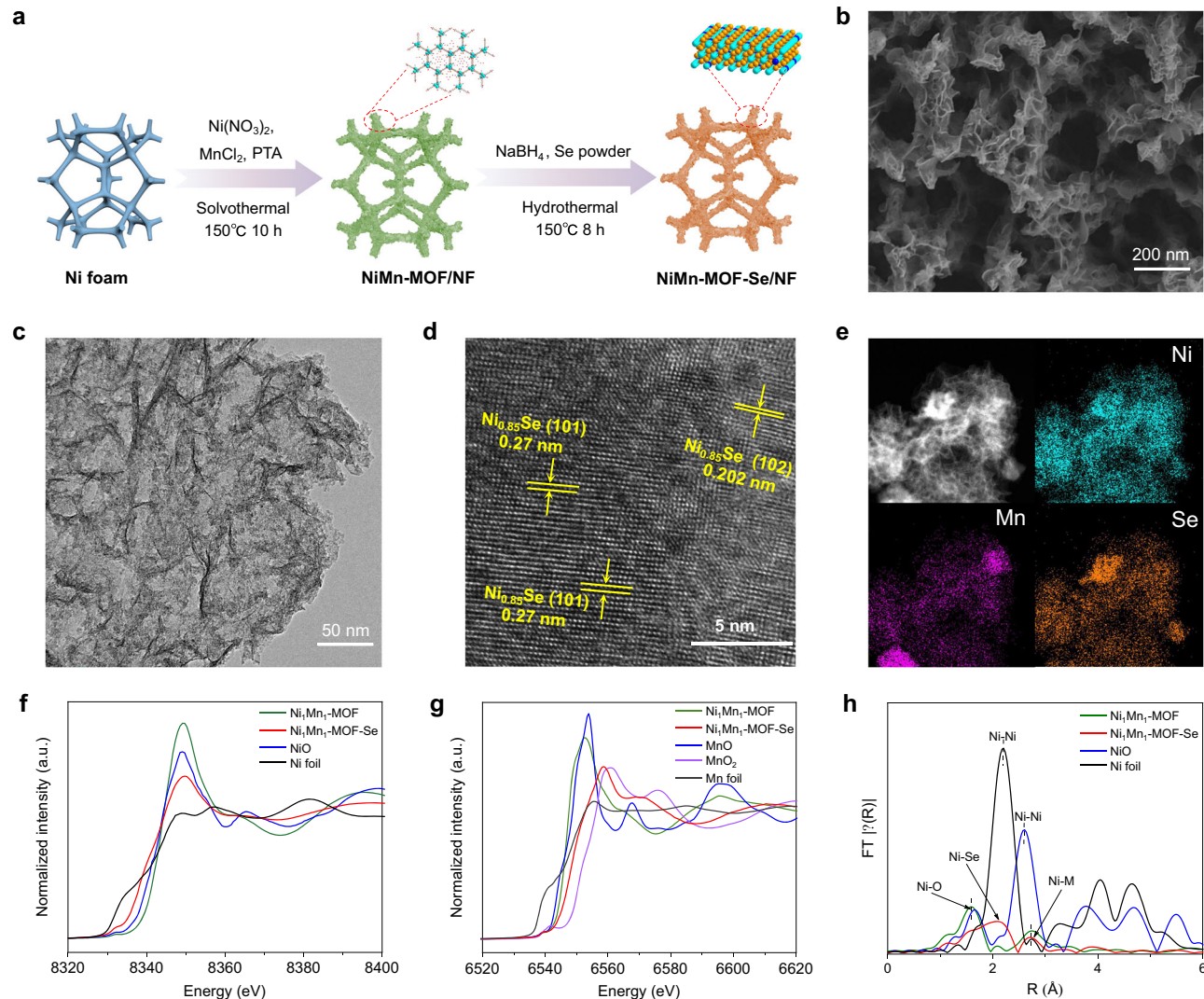

**Fig. 2 | Morphological and structural characterizations of Ni₁Mn₁-MOF-Se/NF anodic catalyst. a** Schematic illustration showing the synthesis of Ni₁Mn₁-MOF-Se/NF. **b** SEM image of Ni₁Mn₁-MOF-Se/NF. **c** TEM image of Ni₁Mn₁-MOF-Se/NF. **d** HRTEM image of Ni₁Mn₁-MOF-Se/NF. **e** STEM-EDX mappings of Ni₁Mn₁-MOF-Se/NF. **f** The

Ni K-edge XANES spectra of Ni₁Mn₁-MOF and Ni₁Mn₁-MOF-Se. **g** The Mn K-edge XANES spectra of Ni₁Mn₁-MOF and Ni₁Mn₁-MOF-Se. **h** The Ni K-edge EXAFS spectra of Ni₁Mn₁-MOF and Ni₁Mn₁-MOF-Se.

(Supplementary Fig. 8), and after electrochemical activation, the Se content decreases from 56.82 wt% to 38.98 wt%, further confirming that the CV activation process led to the substitution of Se by O (Supplementary Table 1). The ICP-OES data of the electrolyte during electrochemical activation process reveals that some of Se and Mn dissolved into the electrolyte. Ni₁Mn₁-MOF-Se/NF experienced some leaching of Mn and Se during the initial test followed by maintaining stable, indicating that Ni₁Mn₁-MOF-Se/NF can work stably during the electrolysis test (Supplementary Table 2).

X-ray photoelectron spectroscopy (XPS) was carried out to investigate the dynamic evolution of surface composition and valence state from Ni₁Mn₁-MOF/NF to Ni₁Mn₁-MOF-Se/NF and the electro-chemically activated Ni₁Mn₁-MOF-Se/NF (Supplementary Fig. 9). For the Ni₁Mn₁-MOF-Se/NF catalyst, the Ni 2$p$ spectrum can be deconvo-luted into eight peaks, including $Ni^0$ (853.8 and 872.1 eV), $Ni^{2+}$ (855.8 and 873.5 eV), $Ni^{3+}$ (857.3 and 875.5 eV), and two satellite peaks[37,42] (Supplementary Fig. 10). Among them, the formation of $Ni^0$ stems from the strong reduction of sodium borohydride. After the electro-chemical activation, the peak corresponding to $Ni^0$ disappears (Sup-plementary Fig. 11) while the peak of $Ni^{3+}$ becomes stronger, suggesting that the activation process facilitates the electronic transition of

surface Ni species. Meanwhile, the XPS results reveals that the surface selenium content decreases from 17.88 wt% to 7.79 wt%, while the oxygen content increases (Supplementary Table 3), confirming the replacement of Se by O once again. This can promotes the formation of NiOOH active species[25]. The Mn element experiences a similar valence state evolution process with that of Ni species.

To get further insights into the influence of high-valence metal Mn and selenization on Ni sites, the electronic structures and the local coordination environments of Ni₁Mn₁-MOF/NF and Ni₁Mn₁-MOF-Se/NF were investigated by the XANES and the extended X-ray absorption fine structure spectra (EXAFS). The Ni K-edge XANES spectra reveal that the absorption threshold of Ni₁Mn₁-MOF/NF shows a shift to higher energy than NiO, indicating an obvious electron transfer from Ni atoms to high-valence Mn[42] (Fig. 2f). Ni₁Mn₁-MOF-Se/NF shows a lower absorption threshold than NiO, implying a lower valance state caused by the weakened electronegativity of Se (2.55) than O (3.44) and the strong reduction of NaBH₄. The Mn K-edge XANES spectra uncover that the valence state of Mn in Ni₁Mn₁-MOF-Se/NF locates at the range between 2+ and 4+ (Fig. 2g). With the application of voltage, surface Se will be gradually replaced by O, and the valence state of Mn will increase to 6+, providing abundant electron holes[42]. Compared

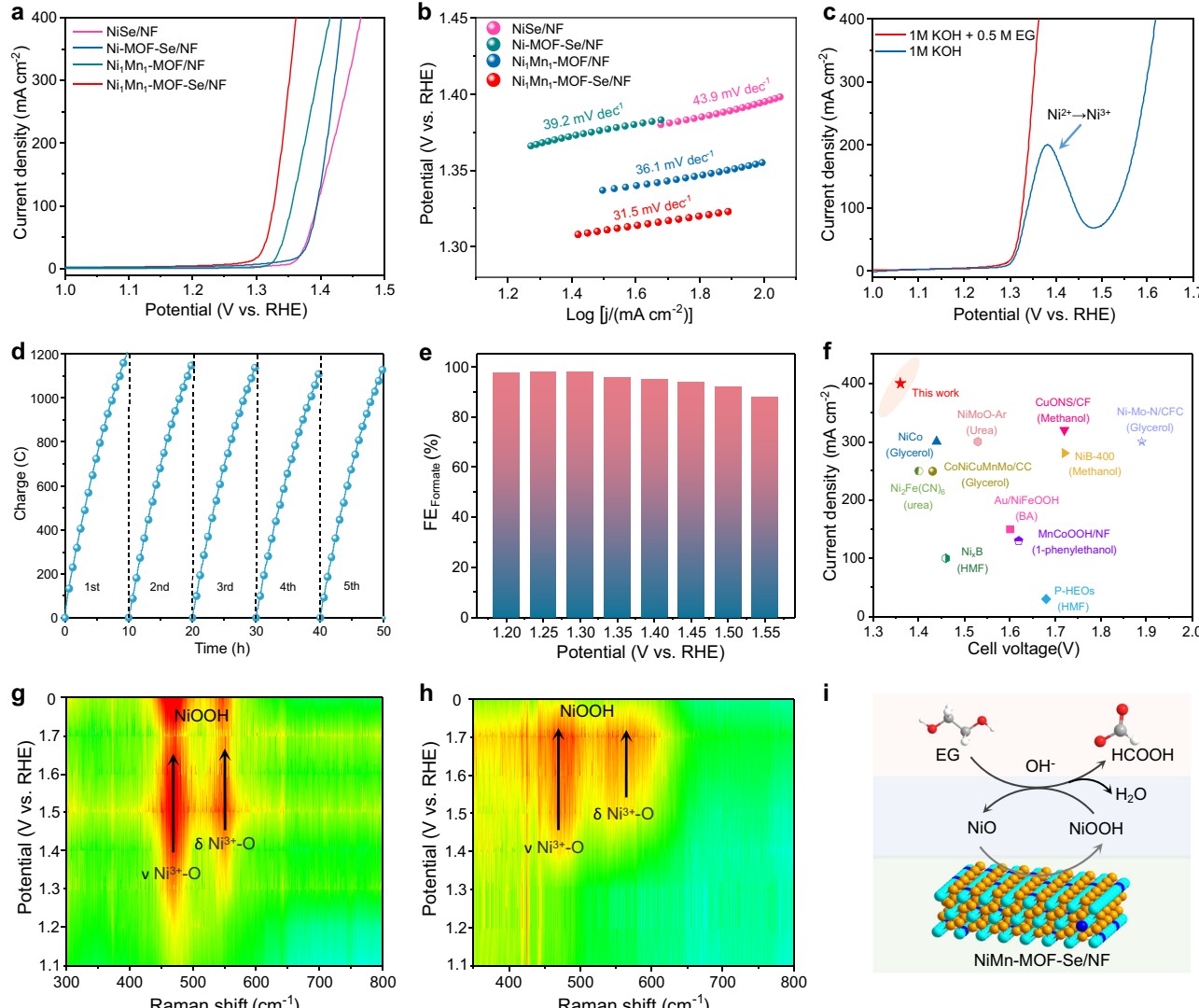

**Fig. 3 | Electrocatalytic performance and mechanistic investigation of Ni₁Mn₁-MOF-Se/NF catalyst toward EOR. a** Linear sweep voltammetry (LSV) curves of different catalysts in 1.0 M KOH with 0.5 M EG (with iR compensation, compensation resistance: 0.9 ± 0.1 Ω). **b** The corresponding Tafel plots. **c** LSV curves comparison of Ni₁Mn₁-MOF-Se/NF in 1.0 M KOH with or without 0.5 M EG. **d** Charge-time curves of Ni₁Mn₁-MOF-Se/NF in catalyzing EOR during the continuous operation of 50 h. **e** FE_formate of Ni₁Mn₁-MOF-Se/NF under different potentials. **f** Comparison of electrolysis performances for AORs between this work and previous works. **g** In situ Raman spectra of Ni₁Mn₁-MOF-Se/NF for OER (1.0 M KOH). **h** In situ Raman spectra of Ni₁Mn₁-MOF-Se/NF for EOR (1.0 M KOH + 0.5 M EG). **i** Schematic illustration of the catalytic reaction mechanism.

with NiO (2.50 Å), the peak position of Ni−Ni/Mn scattering for Ni₁Mn₁-MOF-Se/NF (2.62 Å) exhibits an apparent right shift due to the electronic interaction between Ni and Mn (Fig. 2h). The peaks at 2.15 and 1.80 Å can be attributed to Ni−Se and Ni−O bonds. The bond length indicates that the first shell layer is filled with Ni−Se/O, and Ni−Ni/Mn belongs to the second shell layer. For the R-space spectra of Mn, two peaks at around 1.86 and 2.70 Å can be signed to the Mn−O and Mn−Mn bond, while the peak at 2.76 Å represents the Mn−Ni bond. The EXAFS spectra further confirm the successful doping of Mn into the lattice of NiSe and Ni₀.₈₅Se (Supplementary Fig. 12). The outermost holes of the high-valence Mn can stabilize NiOOH[40,43], while the introduced Se increases the bond length of Ni-OH, synergistically regulating the electronic structure of Ni centers.

### Anodic EG oxidation catalyzed by Ni₁Mn₁-MOF-Se/NF

PET plastic can be depolymerized into PTA and EG monomers in an alkaline medium. EG serves as the substrate for electrochemical oxidation, while PTA does not participate in electronic transfer during the entire electrolytic process due to its unique dual carboxyl structure.

This enables the simplification of the performance evaluation experiment, where 1 M KOH containing 0.5 M EG was used as electrolyte. Ni₁Mn₁-MOF-Se/NF exhibits an unprecedented catalytic activity for EOR under conditions of both with and without *iR* compensation (Fig. 3a, Supplementary Fig. 13). The EOR can easily proceed at industry-level current densities of 100, 200, and 400 mA cm⁻² using ultra-low operating potentials of only 1.327, 1.341, and 1.362 V, outperforming previously reported AORs catalysts (Supplementary Table 4) and the state-of-art OER catalysts[44–46]. This is because of the synergistic effect of high-valence metal Mn and selenium that the electron cloud rearrangement around Ni species promotes the surface electron delocalization and the formation of NiOOH active species, while showing moderate adsorption for intermediates as well. Further exploration toward the optimal proportion of Ni and Mn was carried out. It was found that Ni₁Mn₁-MOF-Se/NF shows a much better catalytic activity for EOR than Ni₁Mn₂-MOF-Se/NF and Ni₂Mn₁-MOF-Se/NF (Supplementary Fig. 14). Tafel curves (Fig. 3b) show that Ni₁Mn₁-Se/NF has a lower Tafel slope (31.5 mV dec⁻¹) than NiSe/NF (43.9 mV dec⁻¹), Ni-MOF-Se/NF (39.2 mV dec⁻¹), and Ni₁M₁-MOF/NF (36.1 mV

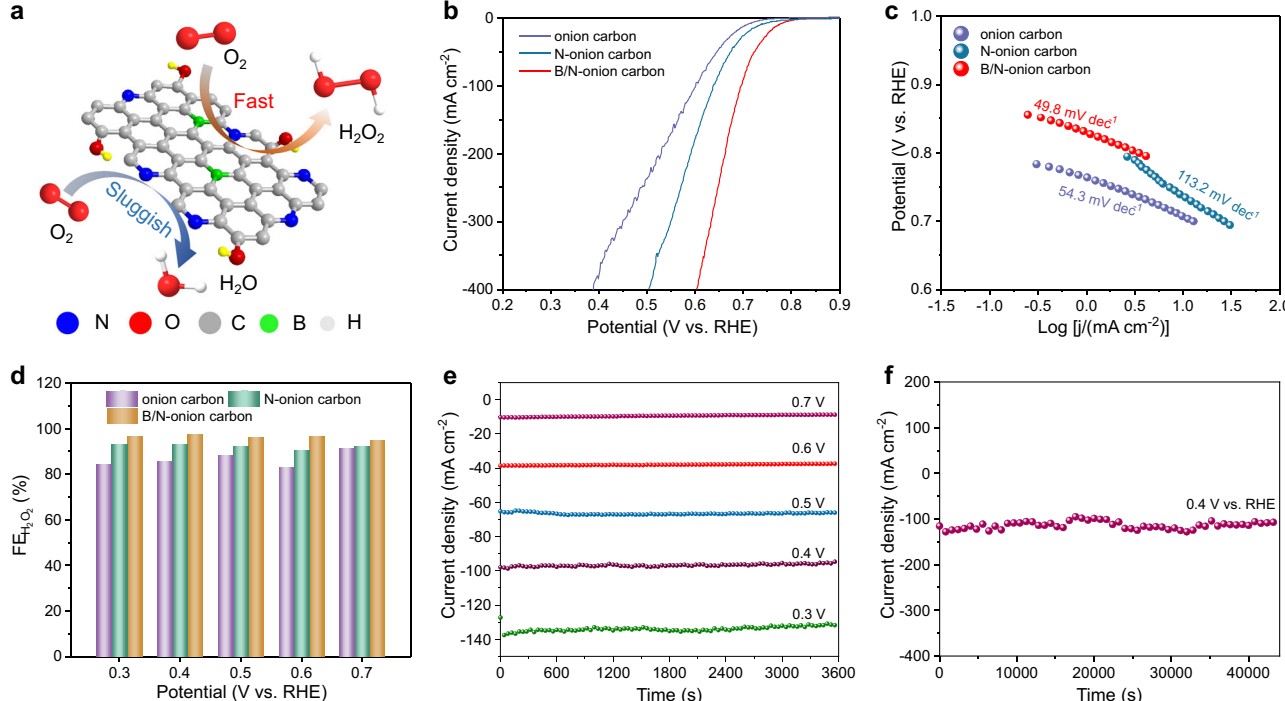

**Fig. 4 | Electrocatalytic performance of B/N-onion carbon toward the ORR-to-H₂O₂ electrosynthesis. a** Schematic diagram of B/N-onion carbon catalyst. **b** LSV curves of different catalysts for ORR in a flow cell (1.0 M NaOH, scan rate of 20 mV s⁻¹, with iR compensation, compensating resistance: 3.1 ± 0.2 Ω). **c** The corresponding Tafel plots. **d** FE$_{H2O2}$ of B/N-onion carbon under different potentials. **e** Current-time curves of B/N-onion carbon under different potentials (without iR compensation). **f** Stability test of B/N-onion carbon at 0.4 V vs. RHE.

dec⁻¹). According to the electrochemical impendence spectra (EIS), Ni₁Mn₁-MOF-Se/NF possesses a lower charge transfer resistance (0.7 Ω) compared to NiSe/NF (8.2 Ω), Ni-MOF-Se/NF (9.3 Ω) and Ni₁Mn₁-MOF/NF (2.5 Ω), benefitting from the excellent conductivity and wettability of the self-supported electrode (Supplementary Fig. 15). Cyclic voltammetry (CV) curves were employed to quantify the electrochemical double-layer capacitance (C$_{dl}$). Ni₁Mn₁-MOF-Se/NF shows a much higher C$_{dl}$ value (22.9 mF cm⁻²) than NiSe/NF (4.8 mF cm⁻²), Ni-MOF-Se/NF (6.8 mF cm⁻²) and Ni₁Mn₁-MOF/NF (5.3 mF cm⁻²) due to the rough surface and array structure expose more activity sites (Supplementary Fig. 16). To demonstrate the energy-saving advantage of EOR over OER, the polarization curves of Ni₁Mn₁-MOF-Se/NF were captured in 1.0 M KOH with and without 0.5 M EG (Fig. 3c). Compared to OER, the more thermodynamically favorable EOR exhibits an apparent decrease of operating potential by 203 mV and 256 mV for reaching the current densities of 100 and 400 mA cm⁻², respectively.

The long-term stability of Ni₁Mn₁-MOF-Se/NF was examined through continuously measuring the transferred charge at a constant voltage of 1.4 V. The electrolyte was refreshed every 10 h, considering that the consumption of the EG substrate may cause the fluctuation of the current response. As shown in Fig. 3d, the catalytic activity of Ni₁Mn₁-MOF-Se/NF approximately shows no degradation during the continuous operation of 50 h. ¹H nuclear magnetic resonance (¹H NMR) spectroscopy was employed to quantitatively detect the product in the electrolyte under different potentials (Fig. 3e, Supplementary Figs. 17 and 18). The electrocatalytic conversion of EG involves the evolution from hydroxy to aldehyde and then to carboxyl, followed by the C−C bond breakage to generate formate[22,32]. Within the operating voltage range of 1.2–1.5 V, the Faradic efficiency of formate (FE$_{formate}$) is maintained at a high value of >90% and the product selectivity is approximately 100% (Fig. 3e). The comparison of current density and cell voltage demonstrates that Ni₁Mn₁-MOF-Se/NF surpasses most previously reported electrocatalysts for small molecule oxidation (Fig. 3f).

## Catalytic mechanism of Ni₁Mn₁-MOF-Se/NF catalyst for EOR

In situ Raman spectroscopy was employed to track the evolution of the Ni²⁺/Ni³⁺ redox pair in Ni₁Mn₁MOF-Se/NF during the EOR with the OER in reference. For the OER, when the applied potential reaches 1.30 V, two signal peaks appear at approximately 473 and 556 cm⁻¹ (Fig. 3g), matching well with the bending vibration (δ(Ni-O)) and stretching vibration (ν(Ni-O)) of Ni³⁺-O of γ-NiOOH[37,47], respectively. With further increasing the voltage, the vibration signals become stronger, which were maintained when removing the voltage input, suggesting an irreversible change of Ni₁Mn₁MOF-Se/NF. The above-mentioned results show that NiOOH is the true active species of OER. The existence of high-valence metal Mn stabilizes NiOOH through bridging Mn−O−Ni bonds, and reduce the adsorption of *OH as well. For EOR with the electrolyte containing 0.5 M EG, however, the vibration signals of γ-NiOOH appear at a higher potential of 1.4 V. In addition, their intensities are apparently weak and sustain unchanged compared with that in the case of OER (Fig. 3h, Supplementary Fig. 19). Moreover, the vibration signals disappear rapidly when terminating the potential input, demonstrating the reversible change of Ni₁Mn₁MOF-Se/NF during the EOR (Fig. 3h). Moreover, the emergence of NiOOH further confirms that the initial Ni₁Mn₁-MOF-Se/NF was attacked by OH⁻ under electric drive, resulting in the substitution of Se by O on the surface. Based on the above-mentioned results and previous works, the EOR mechanism of Ni₁Mn₁-MOF-Se/NF is proposed as follows (Fig. 3i): (1) driven by the anodic potential, Se on NiSe surface is replaced by O to form NiOOH; (2) OH⁻ in electrolyte is adsorbed on electrode surface; (3) the substrate molecule is adsorbed on the surface of the catalyst through the strong interaction between its hydroxyl oxygen and active Ni center; (4) through the nucleophiles attack, the adsorbed EG loses electrons and undergoes dehydrogenation and oxidation, and is then gradually oxidized from hydroxy group to aldehyde group and finally to the carboxyl group. The unstable nature of the double carboxyl group structure causes the breakage of the C−C bond to give formate; (5) The adsorbed formate leaves the electrode surface and dissolves into the electrolyte.

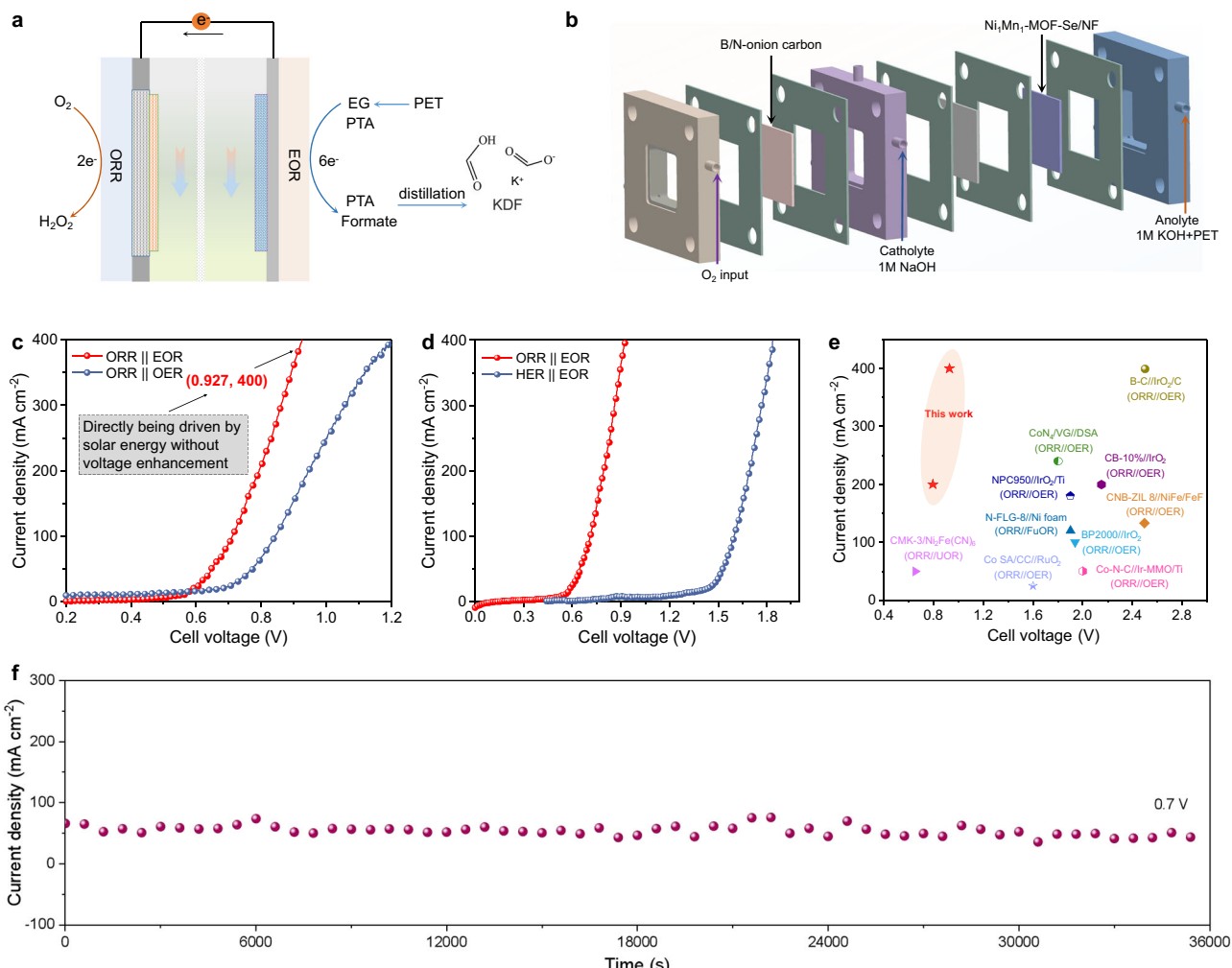

**Fig. 5 | Electrocatalytic performance of the electrosynthesis system electrochemically pairing cathodic ORR-to-$H_2O_2$ with anodic PET upcycling.** **a** Schematic illustration of the electrosynthesis system. **b** Schematic diagram showing the configuration with $Ni_1Mn_1$-MOF-Se/NF and B/N-onion carbon as cathodic and anodic catalysts. **c** Comparison of the polarization curves between the ORR ‖ EOR system and the ORR ‖ OER system (with iR compensation, compensating resistance: $5.6 \pm 0.3 \, \Omega$). **d** Comparison of the polarization curves between the ORR ‖ EOR system and the HER ‖ EOR system. **e** Comparison of cell voltage and current density between this work and previous $H_2O_2$ electrosynthesis systems. **f** Stability test of the ORR ‖ EOR electrolysis system at 0.7 V.

## ORR-to-$H_2O_2$ electrosynthesis catalyzed by B/N-onion carbon

The B, N co-doped onion carbon (B/N-onion carbon) catalyst was prepared through annealing the purchased onion carbon with borax in $NH_3$ atmosphere (Fig. 4a, Supplementary Fig. 20). SEM images show that the ball-like morphology of B/N-onion carbon with a size of 1–2 µm (Supplementary Figs. 21 and 22). Its rougher surface than the original onion carbon promotes the exposure of active sites and surface wettability. The corresponding element mapping exhibits the uniform distribution of B and N, and their doping amounts are 5.1% and 11.0% (atomic ratio) respectively. Nitrogen adsorption-desorption experiment further reveals that the introduction of B and N increased the specific surface area of onion carbon (from 2.9 to 14.0 $m^2 \, g^{-1}$), thereby exposing more active site and improving catalytic activity. B/N onion carbon mainly exhibits the mesoporous structure with an average pore size of 16.3 nm (Supplementary Fig. 23). The Raman spectrum reveals the influence of B/N doping on the defect degree of carbon skeleton (Supplementary Fig. 24). The increased $I_D/I_G$ ratio means that the doping of B and N enriches the defects in onion carbon as the active sites and promotes the electron delocalization. According to the XPS results (Supplementary Figs. 25 and 26), the C 1s spectrum of B/N-onion carbon reveals that the peaks at 283.5, 285.6, and 286.5 eV can be attributed to C–B, C–N, and C–O bonds, suggesting that B and N were

successfully introduced into the carbon materials[9,38]. In the N 1s spectrum of B/N-onion carbon, four peaks can be divided into N–B bond (397.6 eV), pyridinic N (398.5 eV), pyrrolic N (399.4 eV), and graphitic N (400.5 eV)[10]. Compared with N-onion carbon, B/N-onion carbon shows the existence of N–B bond other than C–N and C–B bonds, which contributes to an increased N content in onion carbon from 5.6% to 24.1 at.%. The B 1s spectrum of B/N-onion carbon also exhibits the formation of B-C (189.2 eV), B-N-$C_2$ (190.4 eV), B-$N_2$-C (190.8 eV), and B-$N_3$ (192.0 eV)[48]. Fourier transformed infrared spectroscopy (FTIR) also confirms the formation of B–N bonds[49] (Supplementary Fig. 27). Overall, B and N were mainly introduced into the carbon network in the form of B–N–C species.

We evaluated the ORR-to-$H_2O_2$ activity of B/N-onion carbon in a flow-cell setup with 1 M KOH as electrolyte. According to the current-voltage curves, N-doped onion carbon (N-onion carbon) shows an enhanced ORR catalytic activity than the original onion carbon, which can be attributed to the formation of graphitic N, pyridinic-N, and pyrrolic N[10,48]. The introduction of N can effectively regulate the electronic distribution of carbon network. Further, the introduction of B promotes the formation of B–N bond, which promotes the adsorption of OOH* intermediates together with the N sites, preventing the dissociation from OOH* to O* and OH*, and

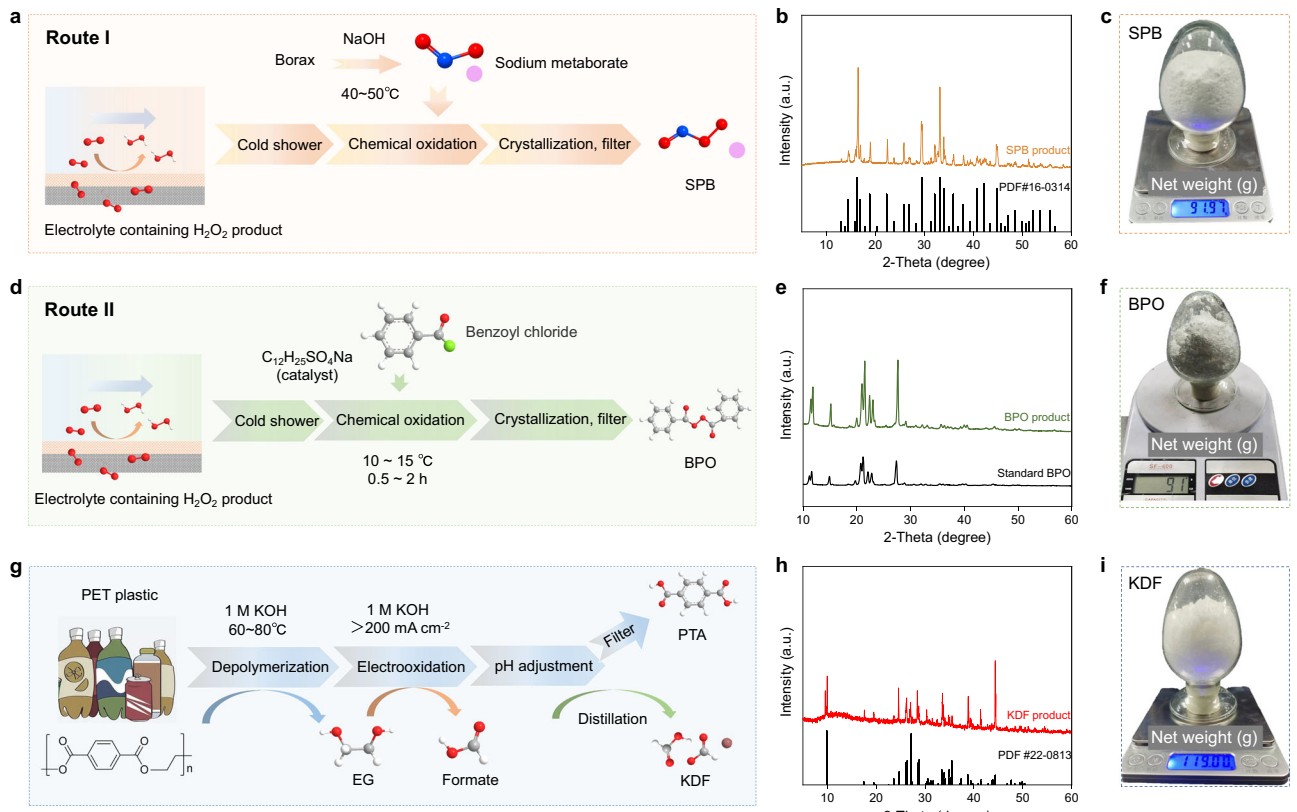

**Fig. 6 | Product upgrade and separation for the energy-saving electrosynthesis.**
**a** Synthesis route of SPB from an electrolyte containing $H_2O_2$. **b** XRD pattern of SPB product. **c** Digital image of SPB product. **d** Synthesis route of BPO from an electrolyte containing $H_2O_2$. **e** XRD pattern of BPO product. **f** Digital image of BPO product. **g** Separation and purification route of anodic products. **h** XRD pattern of KDF product. **i** Digital image of KDF product.

improving the selectivity of $H_2O_2$[9,10,38,48,50,51]. B/N-onion carbon shows outstanding catalytic activity as quantified by the highest starting potential of 0.8 V. It only needs 0.66 and 0.60 V to drive the industrial-scale current densities of 200 and 400 mA cm$^{-2}$ with the *iR* compensation (Fig. 4b), outperforming most of the carbon-based catalysts currently reported (Supplementary Table 5). This superior catalysis performance was also confirmed under the absence of *iR* compensation (Supplementary Fig. 28). B/N-onion carbon catalyst has a lower Tafel slope (49.8 mV dec$^{-1}$) than N-onion carbon catalyst (113.2 mV dec$^{-1}$) and onion carbon (54.3 mV dec$^{-1}$), suggesting that the introduction of B and N can significantly improve the reaction kinetics (Fig. 4c). The FE of $H_2O_2$ (FE$_{H2O2}$) was measured by the potassium permanganate solution titration. As displayed in Fig. 4d, B/N-onion carbon achieves a high FE$_{H2O2}$ of >95% within the whole voltage range of 0.3 - 0.7 V, surpassing the other two catalysts. Especially, the highest FE$_{H2O2}$ reaches 97.5% at the potential of 0.4 V, fully meeting the industrial requirement. A high $H_2O_2$ production rate of 2.48 mmol cm$^{-2}$ h$^{-1}$ can be achieved at a $H_2O_2$ partial current density of 133.0 mA cm$^{-2}$ (Supplementary Fig. 29). The stability of catalyst was confirmed by recording the current densities under different potentials with and without *iR* compensation (Fig. 4e, f, Supplementary Fig. 30). Moreover, the potential-time curve at a constant current density of 100 mA cm$^{-2}$ also confirms the outstanding stability of B/N-onion carbon for ORR-to-$H_2O_2$ (Supplementary Fig. 31). Therefore, the as-developed B/N-onion carbon is a promising catalyst for the scalable ORR-to-$H_2O_2$.

## Proof of concept for electrochemically pairing cathodic ORR-to-$H_2O_2$ with anodic PET upcycling

Electrochemically pairing cathodic ORR-to-$H_2O_2$ with anodic PET upcycling is an effective approach to achieve energy-saving

electrosynthesis due to the lower theoretical potential (1.23 V) called for driving the 2e$^-$ transfer of ORR-to-$H_2O_2$ than HER. We thereby designed the proof-of-concept configuration and examined the performance (Fig. 5a). B/N-onion carbon loaded on the gas diffusion electrode and Ni$_1$Mn$_1$-MOF-Se/NF were employed as cathode and anode, respectively (Fig. 5b). 1 M NaOH and 1 M KOH containing 0.5 M EG were selected as catholyte and anolyte, which are separated by a commercial ion exchange membrane. As predicted, this ORR‖EOR electrosynthesis system successfully worked at the current densities of 100, 200, and 400 mA cm$^{-2}$ with low potential inputs of only 0.712, 0.794, and 0.927 V, surpassing all previously reported $H_2O_2$ electrosynthesis systems[37,52,53] (Fig. 5c and Supplementary Table 6). This low potential requirement of <1 V means that the as-designed electrosynthesis system can be directly powered by the commercial solar cell without using a potential transformer. This excellent performance highlights the thermodynamics and dynamics advantages of EOR compared to OER. To directly show the energy-saving potential, we conducted the baseline experiments with different electrode reactions. On the one hand, the anodic EOR was changed to OER, which suffers from an increase of cell voltage by 270 mV at 400 mA cm$^{-2}$ (Fig. 5c). On the other hand, the cathodic ORR was replaced by HER, leading to an apparent increase of cell voltages by 899 and 910 mV at 200 and 400 mA cm$^{-2}$ respectively (Fig. 5d and Supplementary Table 7). The comparison of current density and cell voltage demonstrates the energy-saving advantage of the as-proposed electrochemistry paring of ORR with EOR over previous $H_2O_2$ electrosynthesis protocols (Fig. 5e and Supplementary Table 6). The stability evaluation of the ORR‖EOR electrosynthesis system shows that it runs stably during the long-term operation of 36,000 s under the cell voltage input of 0.7 V, demonstrating the high possibility for the practical application (Fig. 5f).

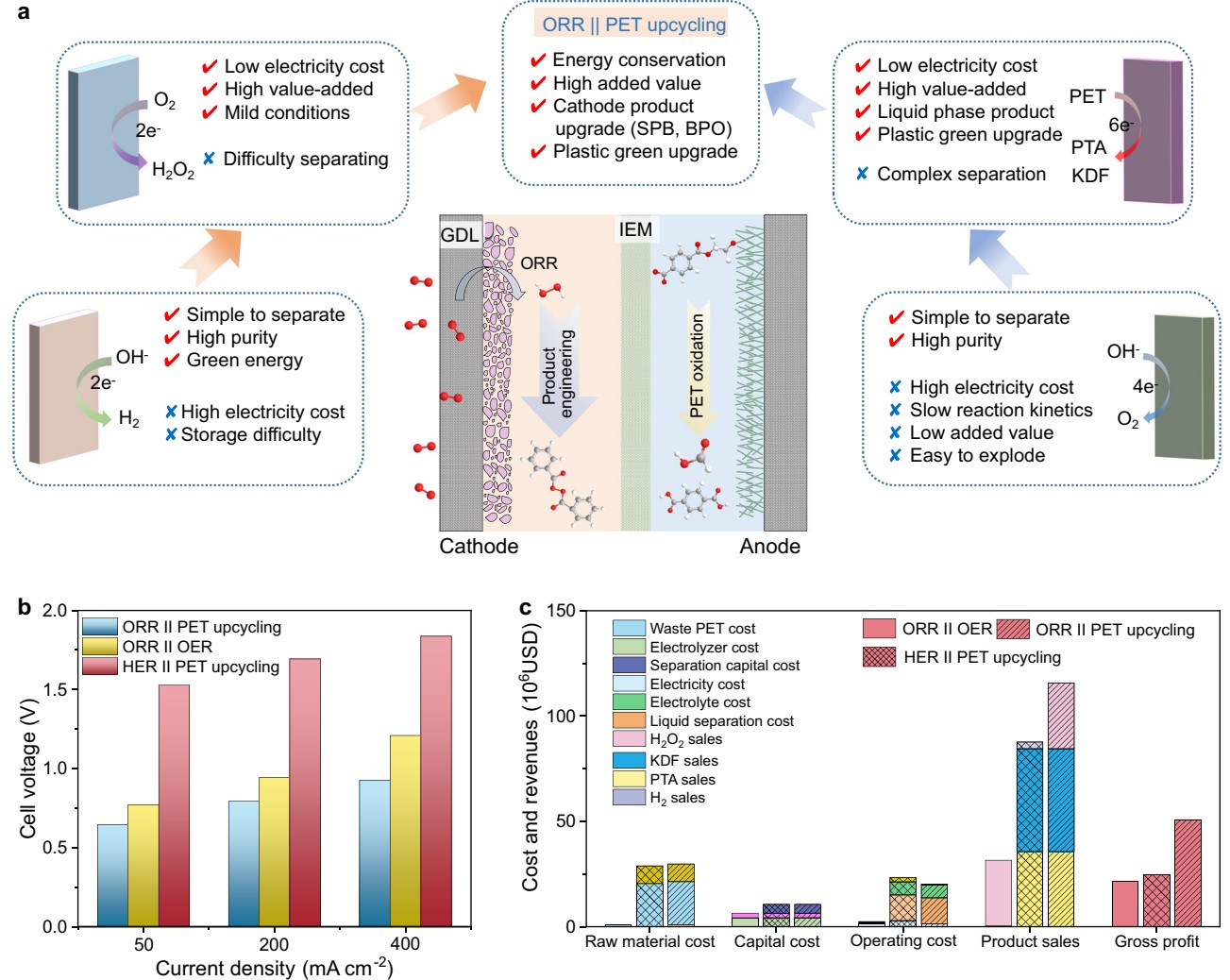

**Fig. 7 | Techno-economic evaluation. a** Comprehensive comparison between electrolysis systems with different catalysis reactions and different product engineering. **b** Energy consumption analysis under different current densities as quantified by cell voltage. **c** Evaluation of economic indicators for different electrosynthesis systems.

## Downstream conversion of H$_2$O$_2$ toward industrial production

Different from the negligible product purification in water splitting, the cost of product separation for H$_2$O$_2$ electrosynthesis even exceeds the operating cost. The economic and efficient product purification strategy plays a significant role in H$_2$O$_2$ electrosynthesis yet has never been reported. Previous researches mainly involve the synthesis of H$_2$O$_2$ while the product was end up with staying in the electrolyte. This is because of the thermodynamic instability of H$_2$O$_2$ in alkaline media under high temperature. To address this problem, we proposed the direct downstream conversion protocols (Supplementary Fig. 32) of H$_2$O$_2$ by upgrading it into value-added chemicals for the first time, including SPB and BPO, which can be synthesized directly in alkaline media. According to our experiment, the concentration of H$_2$O$_2$ influences the yield of downstream products, and the concentration should be more than 5 wt%. Similar with H$_2$O$_2$, SPB can be used as an oxidant, dye-stripping agent, deodorant, and electroplating solution additive. The detailed direct downstream conversion process is shown in Fig. 6a, where borax is dissolved in sodium hydroxide solution in a certain proportion to obtain sodium metaborate. It then reacts with H$_2$O$_2$ in the alkaline electrolytes in an ice water bath to produce SPB. The generated SPB can immediately crystallize and precipitate without the need for subsequent complex purification procedures. The XRD and FTIR

results confirm the successful separation of SPB with a mass of more than 90 g (Fig. 6b, c and Supplementary Fig. 33).

Another direct downstream conversion protocol is to upgrade H$_2$O$_2$ into BPO, which is a widely used initiator in the adhesive industry, e.g., polyvinyl chloride, polyacrylonitrile, acrylic ester, chloroprene rubber, graft polymerization of SBS and methyl methacrylate, and unsaturated polyester resin curing, etc. As shown in Fig. 6d, H$_2$O$_2$ in the electrolyte from the electrosynthesis system could react with benzoyl chloride at 10–15 °C under the catalysis of sodium dodecyl benzene sulfonate to generate BPO, which could be obtained with high yield and purity after filtration, washing, centrifugation, and drying. The XRD and FTIR results confirm the feasibility of the BPO synthetic route (Fig. 6e, f and Supplementary Fig. 34). The as-developed downstream product conversion routes under the regime of H$_2$O$_2$ electrosynthesis process can provide a technical supplement to the oxidant field.

The separation and purification of the anodic product is also worthy of attention. The anolyte after EOR mainly involves the purification of PTA and formate. In view of the poor solubility of PTA in acidic media, the pH of the anolyte was adjusted to 3 - 4 with the help of formic acid (Fig. 6g), obtaining PTA after the crystallization, sedimentation, filtration, washing, and drying processes. The XRD and FTIR results confirm the technical feasibility of upgrading waste PET plastic into PTA and KDF (Fig. 6h, i and Supplementary Figs. 35 and 36).

## Techno-economic evaluation

Technical-economic analysis was conducted to evaluate the practicality of the electrosynthesis process. The technical-economic evaluation covers the fixed investment, the energy consumption of the electrolysis system, the product separation system, and market sales[54,55]. As displayed in Fig. 7a, the standard potential of ORR (1.23 V) is much higher than that of HER (0 V), and the oxidation potentials of AORs (<1 V) are significantly lower than that of OER (1.23 V). As for electricity consumption, the ORR ‖ PET upcycling configuration shows a lower operating voltage compared with the HER ‖ PET upcycling and the ORR ‖ OER configuration. From the perspective of product economic value, the upgrade from waste PET to PTA and formate far exceeds the low-grade $O_2$, realizing the upcycling of waste plastics and meanwhile improving the economic competitiveness of the electrolysis system. In addition, considering the unstable and separation problems of $H_2O_2$ in the alkaline medium, this work proposed two economically feasible direct downstream routes to transform it into BPO and SPB from the perspective of product engineering, which avoids the problem of separation, saves operating costs, and realizes value improvement.

We further demonstrated the economic potential of the ORR ‖ PET upcycling system through quantifying the cost and profit of raw materials[9,32], equipment[17], operating[28], and products (Supplementary Tables 8 and 9), the details of which are provided in the Supplementary Information. Based on the results of the techno-economic evaluation, compared with the HER ‖ PET upcycling configuration and the ORR ‖ OER configuration, the electricity cost of the ORR ‖ PET electrosynthesis system can be reduced by 51.3% and 18.5%, respectively (Fig. 7b). Considering the direct crystallization and precipitation of the cathodic product via upgrading $H_2O_2$ into SPB or BPO, the separation cost of $H_2O_2$ can be exempted. Consequently, the ORR ‖ PET upcycling electrosynthesis system has a higher gross profit of up to $50.8 \times 10^6$ USD, corresponding to an apparent increase of 104.0% than the HER ‖ PET upcycling system ($24.9 \times 10^6$ USD). Moreover, the gross profit achieves a remarkable growth of 134.1% than the conventional ORR ‖ OER system ($21.7 \times 10^6$ USD), highlighting the market competitiveness and economic potential of the ORR ‖ PET electrosynthesis system.

## Discussion

In conclusion, an electrosynthesis system coupling cathodic ORR-to-$H_2O_2$ with anodic waste PET upcycling was developed for the first time. The $Ni_1Mn_1$-MOF-Se/NF catalyst constructed by introducing high-valence metal Mn and heteroatom Se exhibits outstanding catalytic ability in EG oxidation as reflected by the low voltages of 1.327 and 1.362 V to achieve 100 and 400 mA cm$^{-2}$. XAS spectroscopy and in situ Raman spectroscopy reveal the catalytic mechanism of NiOOH as an active species and the synergistic effect of Mn and Se. For cathodic reaction, the designed B/N-onion carbon catalyst for ORR-to-$H_2O_2$ shows the highest FE of 97.5%, and only needs 0.66 V and 0.60 V to drive the current densities of 200 and 400 mA cm$^{-2}$. The ORR ‖ PET upcycling electrosynthesis system only requires super-low cell voltages of 0.712, 0.794, and 0.927 V to achieve the industrial-scale current densities of 100, 200, and 400 mA cm$^{-2}$, outperforming all reported $H_2O_2$ electrosynthesis systems. Moreover, the thermodynamically unstable $H_2O_2$ product in electrolyte was easily upgraded via developing two downstream routes converting $H_2O_2$ to BPO and SPB, which can be separated in a low-energy-consumption and low-cost way. Techno-economic evaluation shows the energy-saving and profit advantages of the ORR ‖ PET upcycling configuration over the HER ‖ PET upcycling configuration and the ORR ‖ OER configuration. This work provides an energy-saving methodology of electrochemistry pairing and product conversion for promoting the industrial application of $H_2O_2$ electrosynthesis.

## Methods

### Materials

Nickel nitrate hexahydrate ($Ni(NO_3)_2 \cdot 6H_2O$, solid, AR), manganese (II) chloride tetrahydrate ($MnCl_2 \cdot 4H_2O$, solid, AR), selenium (Se, solid, 99.9%), sodium borohydride ($NaBH_4$, solid, 98%), boric acid ($H_3BO_3$, solid, 99.5%), benzoyl chloride ($C_7H_5ClO$, liquid, 99%, AR), and potassium hydroxide (KOH, solid, 95%, AR) were purchased from Shanghai Macklin Biochemical Co., Ltd, China. Salicylic acid ($C_7H_6O_3$, solid, 99%, AR) and p-phthalic acid ($C_8H_6O_4$, solid, 99%, AR) were purchased from Aladdin. Formic acid (HCOOH, liquid, 98%, AR), N,N-dimethylformamide (DMF, $C_3H_7NO$, liquid, 99.9%, AR), and ethanol ($C_2H_5OH$, liquid, 99.5%, AR) were purchased from Sinopharm Chemical Reagent Co. Ltd (Beijing, China).

### Synthesis of $Ni_1Mn_1$-MOF-Se/NF

$Ni_1Mn_1$-MOF precursor was constructed on Ni foam ($Ni_1Mn_1$-MOF/NF) through a solvothermal reaction. Specifically, 291 mg of $Ni(NO_3)_2 \cdot 6H_2O$, 198 mg of $MnCl_2 \cdot 4H_2O$, 75 mg p-phthalic acid, and 30 mg salicylic acid were dispersed in a mixed solution of 15 mL DMF, 5 mL ethanol, and 10 mL deionized (DI) water, followed by stirring for 20 min to form a homogeneous solution. Then, the above solution was transferred into a Teflon-lined stainless steel autoclave with a piece of NF (2 cm × 4 cm) at 150 °C for 8 h, followed by naturally cooled down to room temperature. Afterward, the $Ni_1Mn_1$-MOF/NF catalyst was washed with deionized water and ethanol several times, and dried at 60 °C. Similar steps were used to complete the subsequent selenization process. 151 mg of Se powder and 100 mg of sodium borohydride were dispersed in 30 mL DI water, and stirred for 20 min under $N_2$ atmosphere, which was then transferred to a Teflon-lined stainless steel autoclave with $Ni_1Mn_1$-MOF/NF at 140 °C for 8 h. After subsequent cooling, washing, and drying, the $Ni_1Mn_1$-MOF-Se/NF catalyst was successfully synthesized. By changing the ratio (mole ratio) of Ni and Mn in the precursor, $Ni_1Mn_1$-MOF-Se/NF, $Ni_1Mn_2$-MOF-Se/NF, $Ni_2Mn_1$-MOF-Se/NF, and Ni-MOF-Se/NF were synthesized in the same method.

### Synthesis of B/N-onion carbon

Commercially purchased onion carbon was directly used as a carbon source, mixed with boric acid in a ratio of 1:5 and placed evenly in a porcelain boat, and annealed at 800 °C for 2 h under an $NH_3$ atmosphere. After cooling, the obtained powder material was ground and kept in a water bath at 80 °C for 6 h to remove ammonia borate.

### Characterization

The surface morphologies of the as-prepared catalysts were measured by SEM on a ZEISS GeminiSEM 300 field emission scanning electron microscope and TEM on JEOL JEM-F200 high resolution (HR-) TEM operated at 200 kV, combined with energy dispersive X-ray spectroscopy (EDX). XRD patterns were recorded using a Panalytical Empyrean diffractometer equipped with using Cu Kα radiation. XPS were performed on a Thermo Scientific K-Alpha X-ray photoelectron spectrometer using Al Kα X-rays as the excitation source. For the calibration of the XPS spectra for onion carbon, sp$^2$ hybrid carbon (C−C, 284.6 eV) was used as the internal standard to calibrate the XPS spectra of C, N, and B elements. The FTIR measurement was performed on a Thermo Scientific Nicolet iS20. Raman measurements were carried out on a Horiba LabRAM HR Evolution for functional group analysis. ICP-OES measurements were performed using a Thermo Fisher iCAP PRO equipped with a solid-state detector. XAFS spectra of Ni K-edge and Mn K-edge were performed with Si (111) crystal monochromators on beamline 1W1B at the Beijing Synchrotron Radiation Facility (BSRF), China. Before the analysis at the beamline, the powder samples obtained by ultrasound were pressed into thin sheets and sealed using Kapton tape film. Spectra of the standard samples and $Ni_1Mn_1$-MOF-Se and $Ni_1Mn_1$-MOF were recorded under transmission mode. The scanning energy range was set from 8140 to 9130 eV with a step-size of

0.5 eV at the near edge for Ni K-edge. The EXAFS spectra of Ni K-edge and standard samples were recorded in transmission mode. The results were processed with the software codes Athena and Artemis for background subtraction, normalization, and energy calibration.

## Electrochemical measurements

Electrochemical workstation (CHI 760E) was employed to perform all oxidation electrochemical experiments in a standard three-electrode configuration with a Hg/HgO as the reference electrode and a graphite rod as the counter electrode. All working electrodes used in this paper have an area of $1 \times 1 \, cm^2$. The potential was converted to a reversible hydrogen electrode (RHE) via a Nernst equation ($E_{RHE} = E_{Hg/Hg/O} + 0.059 \times pH + 0.098$). EIS measurements were performed in a frequency range from 100,000 to 0.01 Hz with 5 mV amplitude. The electrochemical double-layer capacitances ($C_{dl}$) of various samples were confirmed by CV in the potential region without faradaic process to calculate the $C_{dl}$. The polarization curves were measured at a scan rate of $10 \, mV \, s^{-1}$. All the curves of EOR were used with 85% iR compensation in 1 M KOH with 0.5 M EG electrolyte. The performance measurements of cathode ORR-to-$H_2O_2$ and hybrid electrosynthesis system were carried out in a flow cell. The oxygen feed rate was set to be 30 standard cubic centimeters per minute (sccm) and the flowrate of electrolyte was fixed at $50 \, mL \, min^{-1}$. The polarization curves of cathode ORR and hybrid electrolysis system were measured under conditions of with 100% iR compensation and without iR compensation. The iR compensation method of the CHI 760E electrochemical workstation adopts the instrument's built-in automatic compensation mode, which can be completed by inputting the open circuit voltage and compensation level. The evaluation of the catalysis performance was conducted under the conditions of with and without iR compensation.

## Product quantification

For anode products, chronoamperometry testing was carried out to determine the products of EG oxidation and calculate the corresponding Faradaic efficiencies. Electrolyte solution at different times and potentials were collected, and then analyzed by nuclear magnetic resonance (NMR) spectrometer. $^1H$ NMR spectra were recorded on an AVANCE III HD 400 instrument (Bruker). In which 500 μL electrolyte was added with 100 μL $D_2O$, dimethyl sulfoxide was used as an internal standard. For cathode products, the potassium permanganate ($KMnO_4$) titration method was used to quantitatively analyze $H_2O_2$ produced by cathode ORR. First, 500 ml $KMnO_4$ solution (0.02 M) was heated for 20 min, and then placed in a brown reagent bottle for two days, followed by filtering and saving. The concentration of the prepared $KMnO_4$ solution was calibrated by the calcium oxalate solution with standard concentration. The cathode electrolyte at different potentials were collected, and then analyzed. 20 mL electrolyte and 10 mL $H_2SO_4$ (3 M) were added into a conical flask and dilute to 60 mL with deionized water. $KMnO_4$ solution was gradually added until the color starts to turn light red. The specific reaction equation is as follows:

$$2KMnO_4 + 5H_2O_2 + 2H_2SO_4 = K_2SO_4 + MnSO_4 + 5O_2 + 2H_2O \quad (1)$$

## In situ Raman tests

In situ Raman measurements were conducted by a micro-Raman spectrometer (Renishaw) under an excitation of 633 nm laser light. The single chamber electrochemical operando Raman Cell was designed by the Beijing Scistar Technology Co., Ltd. In detail, catalyst samples, Pt wire, and Ag/AgCl electrode were employed as the working electrode, counter electrode and reference electrode respectively. A series of in situ Raman spectra of the working electrode were collected under chronoamperometry (i-t) at different potentials in a 1.0 M KOH

electrolyte with or without 0.5 M EG. To obtain a clear analysis, the background peak of the window plate was subtracted.

## Data availability

The origin data generated in this study are provided in the Source data file. Additional data are available from the corresponding authors upon reasonable request. Source data are provided with this paper.

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

## Acknowledgements

The authors acknowledge funding support from the National Key R&D Program of China (2022YFB4101600 (J.Q.)), the National Natural Science Foundation of China (U2003216, 22209007 (J.Q.)), the Fundamental Research Funds for the Central Universities, China (buctrc202029, buctrc202129 (J.Q.)), the Beijing Nova Program (Z211100002121093 (Q.Y.)), and the open research fund of Songshan Lake Materials Laboratory (2022SLABFN21 (Q.Y.)). The authors thank Lirong Zheng (Institute of High Energy Physics, Chinese Academy of Sciences, Beijing, 100049 P.R. China) for conducting the X-ray absorption spectroscopy measurements.

## Author contributions

J.Qiu and Q.Y. supervised the project. Q.Y. and J.Qi conceived the idea and carried out the experiments. J.Qi wrote the paper. Q.Y. and J.Qiu revised the paper. Y.D., J.L., and X.Z. helped to conduct parts of the material characterization. N.J., Y.M., and Y.J.M. helped to conduct electrochemical performance test.

## Competing interests

The authors declare no competing interests.
