## [Peer Review File · Nature Communications]

REVIEWER COMMENTS

Reviewer #1 (Remarks to the Author):

The paper by Qiu and Yang reports the development of an electrochemical system able to produce hydrogen peroxide at the cathode throughout a 2e⁻ ORR process and formate at the anode by glycerol electrooxidation. The topic is timely and the paper holds the merit to be the first example of coupling of these two electrochemical processes. At a first glance, the catalytic performance is quite interesting but some critical points outlined below make this reviewer doubtful about the urgency of these results to be published on Nat. Commun.

- the chemical nature of B-N doped materials is not clear and should be investigated in depth. The comment to B 1s and N1s XPS spectra of B/N-codoped onions is too generic and does not add any significative information (“According to the XPS results, B and N were introduced into the carbon network at the form of B-C and C-N bonds”). Moreover, it seems that the calibration of the spectra is different on Suppl Fig 22 and 23 (i.e., C=C reference signal is at different BE), please check it. It is not specified in the experimental part how the spectra have been calibrated. The fitting should also be revised, the N1s component on fig 23 is too large and C1s XPS on Supplementary Figure 22 should be revised (the BE value of C-O component cannot be at lower BE value with respect to the C-N component).

- no information about porosity (SSA and pore size distribution) are provided even if these parameters have pivotal importance on the ultimate electrochemical performance of these materials

- on which basis the ORR activity of N-onions is attributed to pyridinic and graphitic N? It is not explained or based on literature references.

- Figs. 4b and 4e are not coherent at all. How is it possible that Potential and Current density values are so much different? Please fix it.

- the “extraction protocols” for hydrogen peroxide proposed by the authors are wrongly defined or even misleading. Authors do not extract hydrogen peroxide from reaction medium but rather they directly transform it into other products (i.e., sodium peroxyborate and dibenzoyl peroxide). I’m not convinced about the importance and market impact of these products with respect to hydrogen peroxide and, in any case, this derivatization path makes the process not an alternative to the anthraquinone ones (as stated by authors in the introduction). Moreover, it makes misleading the title of the paper where these authors underline the “extraction” of electrochemically produced hydrogen peroxide. Please fix these concepts and rephrase the relative sections.

Overall, I do not feel that the paper in its current form deserves publication in Nat. Commun. A more specialized journal (i.e, ChemElectroChem) would be a better vehicle for this piece of chemistry proviso that all major points listed above are properly addressed in a revised document.

PS. Please note that some figures available in the source files are missing in the PDF version of the Supporting Material.

Reviewer #2 (Remarks to the Author):

The electrosynthesis of H₂O₂ is attracting worldwide attention, whereas the practical application of this booming field still suffers from challenges such as the difficult extraction of the thermodynamically unstable H₂O₂ product, the high anodic potential of oxygen evolution reaction, and the low-value product of oxygen. In this manuscript, the authors reported an energy-saving protocol for H₂O₂ electrosynthesis through the electrochemistry pairing and the easy extraction of H₂O₂ product. The as-reported electrolysis system run successfully at an industrial-scale current density of 400 mA cm⁻² using a super-low cell voltage (0.927 V). Moreover, the H₂O₂ electrosynthesis process was push forward from current synthesis step to the downstream product extraction by converting H₂O₂ to BPO and SPB, which represents a substantial advance of this booming field. Thus, I would like to recommend the publication of this work after a minor revision.

1. More detailed descriptions need to be provided regarding the coordination environment of the NiMn-MOF-Se/NF catalyst and the synergistic regulation mechanism of Se and Mn, especially the analysis of X-ray absorption spectroscopy and in-situ Raman spectra.

2. The morphology of Ni₁Mn₁-MOF-Se/NF after electrochemical activation (Fig. S5) is very different from the fresh Ni₁Mn₁-MOF-Se/NF (Fig. S4), such a large surface reconstruction will affect the electrochemical performance of catalysts, please discuss this issue. In addition, almost all the Se has been leached after electrochemical activation, the Ni₁Mn₁-MOF-Se/NF is actual as a precatalyst, the role of Se and the true active catalyst should be discussed.

3. The co-doping of B and N elements in onion carbon was examined to play a significant role in enhancing the performance of H₂O₂ electrosynthesis. The authors are suggested to provide more discussions about the synergetic mechanism between B and N. Additionally, please provide the partial current densities and yield of H₂O₂ at different potentials.

4. The two downstream extraction protocols proposed in this manuscript (through directly upgrading H₂O₂ into sodium perborate and dibenzoyl peroxide) are meaningful for H₂O₂ electrosynthesis. Why does the authors choose these two downstream products? Is there any other possible downstream product?

5. In the economic evaluation, why did the authors not consider the separation cost from H₂O₂ to downstream products? Will the concentration of H₂O₂ in electrolyte affect the yield rate of downstream products.

6. In this manuscript, the authors evaluated the ORR-to-H₂O₂ activity and the performance of the electrolysis system in a flow-cell setup. Please provide discussions about the underlying challenges and future directions from the perspective of the practical use.

7. Several figures are missing in the supporting information.

Reviewer #3 (Remarks to the Author):

This work shows an original electrochemical cell producing at the cathodic side H₂O₂ and at the anode formate from PET degradation. The Ni-Mn bimetal- and onion carbon-based catalysts designed for driving the two reactions work effectively and enable achieving high current density at modest cell potential. The work is further enhanced by a techno-economic analysis regarding the novel proposed electrosynthesis process. Overall the quality and novelty of this work are very high, ensuring this manuscript a cutting-edge character that will help the development of the field. Therefore, I recommend its publication after minor revisions as noted below.

The EDS mapping of the various electrocatalysts show that Ni is homogeneously present in the materials while Se and Mn accumulated in some regions. Can the authors comment on this and add a discussion about this point?

The authors report in the SI (Figure S8) the ICP-OES results of the Ni₁Mn₁-MOF-Se/NF catalyst before and after activation. It would be insightful to add the ICP of the solution after electrocatalysis tests to see if the catalysts experienced any leaching.

The following references can be added as relevant examples of H₂O₂ electrosynthesis with N-doped carbon nanostructures.

Enhanced On-Site Hydrogen Peroxide Electrosynthesis by a Selectively Carboxylated N-Doped Graphene Catalyst

ChemCatChem 13 (20), 4372-4383 (2021)

N-doped graphitized carbon nanohorns as a forefront electrocatalyst in highly selective O₂ reduction to H₂O₂

Chem 4 (1), 106-123 (2018)

Reviewer #4 (Remarks to the Author):

The authors report on the electrocatalytic production of H₂O₂ coupled with the oxidation of ethylene glycol on C- and NiMnSe-based electrodes. They also discuss the extraction of H₂O₂ and present an economical study of the process. The work is well completed, going from the material synthesis and characterization to the device test and economic evaluation and the reported process is of interest. However, the manuscript contains numerous errors and imprecisions that decrease its credibility and must be corrected before further evaluation:

1. The authors state that they couple H₂O₂ production with PET upcycling but H₂O₂ generation reaction is coupled with the ethylene glycol oxidation. While ethylene glycol can be produced from PET, it is misleading to state that H₂O₂ is coupled with PET recycling.
2. References for all the cost evaluations need to be provided. References for values in Table 6 must be also provided. It is not clear if the cost of all the chemicals (e.g. formic acid) considered in the calculation
3. During the activation of the catalyst, all the Se is lost according to the supplementary Figure 5. The authors should discuss this point. The authors should also discuss what the activation process consists of.
4. EDX data should be provided with less significant digits.
5. In supplementary figure 9a, an XPS peak is missing at about 878 eV. Why the intensity of the 2p_{1/2} peaks of Mn is so low, this has no physical sense. Mn²⁺ and NiO should provide doublets in the XPS spectra, but only the 2p_{3/2} peak is fitted.
6. Some references are misplaced (or do not refer to the text where they are inserted)
7. In figure 1a, H₂O should be H₂O₂

Response to Reviewers

Dear Reviewers:

Thanks a lot for your constructive comments and suggestions about our manuscript entitled “Energy-saving and product-oriented H₂O₂ electrosynthesis: from electrochemistry pairing to product extraction” (Manuscript number: NCOMMS-23-17553-T). Those comments are all valuable and very helpful for improving the quality of our paper. We have studied your comments very carefully and substantially revised our manuscript in hope of addressing your concerns and meeting the high standards of *Nature Communications*. The comments were addressed point-by-point below and the involved changes have been highlighted in the revised manuscript and supporting information.

To Reviewer #1:

Comment 1: The chemical nature of B-N doped materials is not clear and should be investigated in depth. The comment to B 1s and N 1s XPS spectra of B/N-codoped onions is too generic and does not add any significative information (“According to the XPS results, B and N were introduced into the carbon network at the form of B-C and C-N bonds”). Moreover, it seems that the calibration of the spectra is different on Suppl Fig 22 and 23 (i.e., C=C reference signal is at different BE), please check it. It is not specified in the experimental part how the spectra have been calibrated. The fitting should also be revised, the N 1s component on fig 23 is too large and C 1s XPS on Supplementary Figure 22 should be revised (the BE value of C-O component cannot be at lower BE value with respect to the C-N component).

Reply:

Thank you for the valuable comments. We have taken your suggestion to calibrate the XPS spectra in Supplementary Figs. 22 and 23 (corresponding to Supplementary Figs. 24 and 25 in the revised version). Considering that onion carbon is mainly composed of carbon with sp² hybrid orbitals, we use the sp² hybrid carbon (C-C, 284.6 eV) to calibrate the XPS spectra. The corresponding description about the calibration of the XPS spectra has been added in the characterization section (page 20 in the revised manuscript).

According to the literatures (*Nat. Catal.* 2022, 5, 1110-1119; *Adv. Energy Mater.* 2020, 1903289; *Adv. Funct. Mater.* 2022, 32, 2113335), we re-fitted the C 1s XPS spectra in Supplementary Figs. 24 and 25, where the peaks at 285.8 and 286.8 eV can be attributed to the C-N and C-O bonds, respectively.

For the N 1s spectrum in Supplementary Fig. 25, we performed XPS characterization of B/N-doped onion carbon again, and the results agree well with the previous results. In the N 1s spectrum of B/N-onion carbon, four peaks can be divided into N-B bond (397.6 eV), pyridinic N (398.5 eV), pyrrolic N (399.4 eV), and graphitic N (400.3 eV) (*Adv. Energy Mater.* 2020, 1903289; *Angew. Chem.* 2022, 134, e202206915). Compared with N-onion carbon, B/N-onion shows the existence of N-B bond other than C-N and C-B bonds, which contributes to an increased N content in onion carbon from 5.6% to 24.1 at%. Therefore, the N 1s component in the B/N-onion carbon appears to be higher than that of N-onion carbon. The B 1s spectrum of B/N-onion carbon also exhibits the formation of B-C (189.2 eV), B-N-C₂ (190.4 eV), B-N₂-C (190.8 eV), and B-N₃ (192.0 eV) (*Angew. Chem.* 2022, 134, e202206915). Overall, B and N were mainly introduced into the carbon materials at the form of B-N-C species (*Nat. Commun.* 2021, 12, 4225). Additionally, we supplemented the Fourier transformed infrared spectroscopy (FTIR) of B/N-doped onion carbon (Supplementary Fig. 26), further confirming the formation of B-N bonds (*Angew. Chem. Int. Ed.* 2022, 61, e202207807).

We have taken your suggestions to add these discussions about the chemical nature of B/N-onion carbon on page 11 in the revised manuscript. The corresponding description about the calibration has been added in the characterization section on page 20. The re-fitted XPS spectra were provided in the revised Supporting Information (Supplementary Figs. 24 and 25). Please check

Comment 2: No information about porosity (SSA and pore size distribution) are provided even if these parameters have pivotal importance on the ultimate electrochemical performance of these materials.

Reply:

Thank you for the insightful comment. Porosity details including the specific surface

area (SSA) and the pore size distribution matter much for the exposure of active site as well as the ion transport. According to your suggestion, we have supplemented the Nitrogen adsorption-desorption isotherms and pore distribution of B/N-onion carbon in the Supporting Information (Supplementary Fig. 22) and on page 11 in the revised manuscript. It can be found that the doping of N and B increases the specific surface, thereby exposing more active site and improving catalytic activity. The pore distribution results indicate that B/N onion carbon mainly exhibits the mesoporous structure with an average pore size of 16.3 nm. We have added these discussions about porosity on page 11 in the revised manuscript. Please check

Comment 3: On which basis the ORR activity of N-onions is attributed to pyridinic and graphitic N? It is not explained or based on literature references.

Reply:

Thank you for the valuable comment and suggestion. According to the literature, graphitic-N and three pyridinic-N sites tend to drive ORR to generate H₂O₂ through 2e⁻ transfer path (*Adv. Energy Mater.* 2021, 11, 2002459; *Adv. Energy Mater.* 2020, 10, 2000789). Additionally, the formation of pyrrolic-N and B-N bonds also promotes the occurrence of 2e⁻ transfer path (*Angew. Chem.* 2022, 134, e202206915). Specifically, these N-based active sites can regulate the charge distribution of the carbon network to promote the adsorption of OOH* intermediate, preventing the dissociation from OOH* to O* and OH* and thus improving the selectivity of H₂O₂ (*Adv. Energy Mater.* 2021, 11, 2002459; *Angew. Chem.* 2022, 134, e202206915).

We have taken your suggestion to add these detailed discussions and the references about the catalytic mechanism of N sites on page 11 in the revised manuscript. Please check

Comment 4: Figs. 4b and 4e are not coherent at all. How is it possible that Potential and Current density values are so much different? Please fix it.

Reply:

Thank you for the insightful comment. The difference about potential is caused by the *iR*-compensation. In Fig. 4b, linear sweep voltammetry tests were conducted with 100%

iR-compensation to eliminate voltage drops caused by solution resistance and liquid connection resistance. This also facilitates the accurate comparison of performance between this work and previously reported catalysts (tested with *iR*-compensation, *Nat. Commun.* 2021, 12, 4225; *Nat. Commun.* 2022, 13, 2880). In Fig. 4e, the current-time curves were recorded using the chronoamperometry tests to measure the selectivity of H₂O₂ at different potentials without *iR*-compensation, and there is no compensation for the solution resistance, resulting in the differences of potential compared with Fig. 4b.

To avoid the misunderstanding about the difference of potential, we have taken your suggestion to add the details on page 12 and 13 in the revised manuscript. Please check

Comment 5: The “extraction protocols” for hydrogen peroxide proposed by the authors are wrongly defined or even misleading. Authors do not extract hydrogen peroxide from reaction medium but rather they directly transform it into other products (i.e., sodium peroxyborate and dibenzoyl peroxide). I’m not convinced about the importance and market impact of these products with respect to hydrogen peroxide and, in any case, this derivatization path makes the process not an alternative to the anthraquinone ones (as stated by authors in the introduction). Moreover, it makes misleading the title of the paper where these authors underline the “extraction” of electrochemically produced hydrogen peroxide. Please fix these concepts and rephrase the relative sections.

Reply:

Thank you for the valuable comments. To avoid the misunderstanding, we have taken your suggestion to replace the “product extraction” with “product conversion” in the manuscript including sections of title, abstract, and the main text as highlighted on pages 1, 3, 14, 15, etc. Downstream product conversion is used to present the characteristic that the product in our work can be easily separated via crystallization and precipitation rather than remaining in the mixture of liquid electrolytes after the electrosynthesis process reported in previous works. H₂O₂ electrosynthesis and product conversion represent the attempt to develop strategy for addressing the challenges encountered by the traditional processes such as the anthraquinone process. However, previous works about H₂O₂ electrosynthesis ended up with the product staying in

electrolytes, which poses the energy-consumption challenge because of the thermodynamic instability of H₂O₂. Thus, we have taken your insightful suggestion to use the description of “product conversion” to avoid the misunderstanding and demonstrate the advantage of our work.

Thanks again for your comment about the importance of sodium peroxyborate and dibenzoyl peroxide. Sodium peroxyborate have been widely used as an oxidant in fields of dye-stripping agent, deodorant, and electroplating solution additive. Because of its good bleaching ability but not damaging the fibers, sodium peroxyborate have been used in protein fibers such as wool, silk, and cotton. In medicine, sodium perborate can also be used as disinfectant and bactericide. As another strong oxidant, dibenzoyl peroxide has been widely used as initiator in polymerization. In 2022, the global hydrogen peroxide market exceeded 4 billion (USD, *J. Am. Chem. Soc.* 2022, 144, 6, 2603-2613), while the industrial demands for two downstream products reach a market size of over 1.5 billion (USD).

The anthraquinone process has been developed for many years and dominates the H₂O₂ market even though facing the challenges such as cost and energy consumption. The as-developed downstream product conversion routes in our work under the regime of H₂O₂ electrosynthesis process just tries to provide a technical supplement, rather than replacing the anthraquinone process.

We have taken your suggestion to replace the “product extraction” by “product conversion” in the revised manuscript. We have also added these discussions about the significance of the products such as sodium peroxyborate and dibenzoyl peroxide as a technical supplement to anthraquinone process (on page 15 in the revised manuscript). Please check

Comment 6: Please note that some figures available in the source files are missing in the PDF version of the Supporting Material.

Reply:

Thank you for the valuable comment. We added all figures in the submitted word file, while some figures are missing in the PDF file. This occurred when converting the

Word file to the PDF file. To this end, we have tried to directly resubmit the revised Supporting Information with the PDF format. Please check

To Reviewer #2:

Comment 1: More detailed descriptions need to be provided regarding the coordination environment of the NiMn-MOF-Se/NF catalyst and the synergistic regulation mechanism of Se and Mn, especially the analysis of X-ray absorption spectroscopy and in-situ Raman spectra.

Reply:

Thank you for the beneficial suggestion. The regulation of the electronic structure of the Ni species is important to improve the intrinsic activity of the catalysts (*Nat. Commun.* 2019, 10, 5335; *Nat. Energy*, 2021, 6, 904-912; *Nat. Commun.* 2022, 13, 3777). Embedding high-valence metals (Mn, V, W, etc.) is an effective strategy to regulate the adsorption strength of 3d metals (Ni, Co, Fe) toward reaction intermediates and the oxidation potential (*Science* 2016, 352, 333-337; *Nat. Catal.* 2020, 3, 985-992). During the electrolytic process, the outermost orbital of high-valence metal Mn with rich holes can realize the transfer of electrons from Ni to Mn, thereby stabilizing NiOOH active species (*Angew. Chem. Int. Ed.* 2021, 60, 10577-10582).

In addition, due to the low electronegativity of Se (2.55) compared to O (3.44), electrons tend to migrate towards the surface through the Se-Ni-O bond, and the larger atomic radius of Se than O can extend the Ni-OH bond length, thereby accelerating the self-loop of Ni²⁺/Ni³⁺ species (*Nat. Commun.* 2022, 13, 2916; *Angew. Chem. Int. Ed.* 2021, 133, 22181-22187). Overall, Mn and Se synergistically regulate the electronic structure of the active center Ni and promote the conversion of Ni-OH to NiOOH. X-ray absorption spectroscopy confirms the coordination environment around Ni and the electronic regulatory effects of Mn and Se on Ni species. In situ Raman spectroscopy further reveals that NiOOH is the true active specie for catalyzing ethylene glycol (EG) oxidation.

The above-mentioned descriptions have been added in the revised manuscript as highlighted on pages 4, 6 and 9. Please check

Comment 2: The morphology of Ni₁Mn₁-MOF-Se/NF after electrochemical activation (Fig. S5) is very different from the fresh Ni₁Mn₁-MOF-Se/NF (Fig. S4), such a large surface reconstruction will affect the electrochemical performance of catalysts, please discuss this issue. In addition, almost all the Se has been leached after electrochemical activation, the Ni₁Mn₁-MOF-Se/NF is actual as a precatalyst, the role of Se and the true active catalyst should be discussed.

Reply:

Thank you for the insightful comment. The electrochemical activation process was completed via the cyclic voltammetry (CV) method from -0.2 to 0.5 V (vs Hg/HgO) in an electrolyte containing 1 M KOH and 0.5 M ethylene glycol (EG). The SEM images (Supplementary Fig. 4) shows that NiMn-MOF-Se/NF has an array structure of flakes. After the CV activation, the reconstructed catalyst evolves into smaller nanosheets with numerous groove structures (Supplementary Fig. 5), which promotes the exposure of active sites and facilitates the mass transfer of electrolyte. CV activation causes the oxidation of Ni and Mn in the NiMn-MOF-Se/NF catalyst, demonstrating the reconstruction on the catalyst surface. This reconstruction can be confirmed by XPS spectra (Supplementary Figs. 9-11). The EDS mapping images and the XPS spectra confirm that the Se element on the surface was gradually replaced by O, which was also reported by literature (*Nat. Commun.* 2022, 13, 2916). This reconstructed catalyst can increase the valence state of surface Ni species, making it easier to convert into the real active species NiOOH, thereby enhancing catalytic activity. We have supplemented these discussions on page 4 in the revised manuscript. Please check

During the activation process, the surface of NiMn-MOF-Se/NF was attacked by the nucleophilic reagent OH⁻ to replace Se by O, while the bulk of NiMn-MOF-Se/NF still maintained the crystal structure of NiSe and Ni_{0.85}Se (Supplementary Fig. 3). Se with large atomic radius can increase the bond length of Ni-OH, thereby promoting the conversion from Ni-OH to NiOOH. At the same time, selenidation can improve the conductivity of the catalyst, thereby promoting the electron transfer. The true active center used for catalytic ethylene glycol (EG) oxidation is the NiOOH species on the catalyst

surface, which has been confirmed by the in situ Raman spectra and previous works (*Nat. Commun.* 2022, 13, 4602; *Nat. Commun.* 2022, 13, 2916). The introduction of Mn and Se has a synergistic effect on regulating the electronic structure of Ni and stabilizing NiOOH species.

We have supplemented these discussions on pages 4, 5 and 10 in the revised manuscript. Please check

Comment 3: The co-doping of B and N elements in onion carbon was examined to play a significant role in enhancing the performance of H₂O₂ electrosynthesis. The authors are suggested to provide more discussions about the synergetic mechanism between B and N. Additionally, please provide the partial current densities and yield of H₂O₂ at different potentials.

Reply:

Thank you for the beneficial suggestion. The N-doping onion carbon can induce abundant defects in carbon microstructures and alter the electron distribution of the carbon network, which can enhance the adsorption of OOH* intermediates, thus promoting the formation of H₂O₂ through 2e⁻ pathway (*Adv. Energy Mater.* 2021, 11, 2002459; *Energy Environ. Sci.*, 2022, 15, 2858). The B doping can also enhance the adsorption of OOH* and help reduce the energy barrier towards 2e⁻ pathway (*Nat. Commun.* 2021, 12, 4225). Additionally, the introduction of B promotes the formation of B-N bonds, thereby increasing the N content in onion carbon from 5.6% to 24.1 at%. Meanwhile, the N 1s XPS spectrum (Supplementary Fig. 25b) indicates that B/N-onion carbon contains abundant N sites and N-B bonds, which can synergistically regulate the charge distribution of the carbon network to promote the adsorption of OOH* intermediate, preventing the dissociation from OOH* to O* and OH*, and thus improving the selectivity of H₂O₂ (*Angew. Chem.* 2022, 134, e202206915).

According to your suggestion, we have added discussions about the synergetic mechanism between B and N on page 11 in the revised manuscript. Additionally, we have taken your suggestion to add the partial current densities and yield of H₂O₂ at different potentials (Supplementary Fig. 27) in the revised Supporting Information. Please check

Comment 4: The two downstream extraction protocols proposed in this manuscript (through directly upgrading H_2O_2 into sodium perborate and dibenzoyl peroxide) are meaningful for H_2O_2 electrosynthesis. Why does the author choose these two downstream products? Is there any other possible downstream product?

Reply:

Thank you for the beneficial comment. There are many downstream products of hydrogen peroxide, such as sodium perborate (SPB), sodium percarbonate (SPC), dibenzoyl peroxide (BPO), dimethyl sulfoxide (DMSO), and calcium peroxide. The main reason for choosing these two downstream products (SPB and BPO) is that their preparation process is conducted in an alkaline media (such as NaOH solution), which means that the electro-synthesized H_2O_2 in alkaline electrolyte can be directly used to produce the downstream products without the need of pre-separation from the electrolyte. The synthesis of other downstream products requires the separation and purification of H_2O_2 , which increases the operating cost.

We have taken your suggestion to add these discussions on page 15 in the revised manuscript. Please check

Comment 5: In the economic evaluation, why did the authors not consider the separation cost from H_2O_2 to downstream products? Will the concentration of H_2O_2 in electrolyte affect the yield rate of downstream products?

Reply:

Thank you for the beneficial comments. The products of sodium perborate (SPB) and dibenzoyl peroxide (BPO) can easily crystallize and participate in the liquid medium due to their low solubility in water, thus exempting the subsequent separation procedure. Therefore, the separation cost can be ignored in the economic evaluation. According to our experiment, the concentration of H_2O_2 influences the yield of downstream products, and the concentration should be more than 5 wt%.

Please check the added discussions as highlighted on page 15 in the revised manuscript.

Comment 6: In this manuscript, the authors evaluated the ORR-to-H₂O₂ activity and the performance of the electrolysis system in a flow-cell setup. Please provide discussions about the underlying challenges and future directions from the perspective of the practical use.

Reply:

Thank you for the beneficial comments. The H₂O₂ electrosynthesis faces the challenges of preparing catalysts with low-cost, scalable preparation, and long-term stability. The scaleup of the H₂O₂ electrosynthesis from laboratory study to industrial application faces the challenges such as the uniform loading of catalysts and the stable operation of devices. Developing anodic small molecule oxidation reactions to replace the oxygen evolution reaction (OER) represents an effective strategy to reduce the cell potential for achieving the energy-saving H₂O₂ electrosynthesis.

These discussions have been added in the revised manuscript on page 2. Please check

Comment 7: Several figures are missing in the supporting information.

Reply:

Thank you for the valuable comment. We added all figures in the submitted word file, while some figures are missing in the PDF file. This occurred when converting the Word file to the PDF file. To this end, we have tried to directly resubmit the revised Supporting Information with the PDF format. Please check

To Reviewer #3:

Comment 1: The EDS mapping of the various electrocatalysts show that Ni is homogeneously present in the materials while Se and Mn accumulated in some regions. Can the authors comment on this and add a discussion about this point?

Reply:

Thank you for the beneficial suggestion. According to the EDS mapping images of NiMn-MOF-Se/NF before activation (Supplementary Figs. 4 and 6), Ni, Mn, and Se appear to be uniformly distributed. The XRD patterns confirm the lattice doping of Mn

and the formation of NiSe and Ni_{0.85}Se. The EDS mapping images of NiMn-MOF-Se/NF after activation (Supplementary Figs. 5 and 7) reveal that Ni still maintains a uniform distribution, while Mn and Se exhibit aggregation to some degree. This is related to the electrochemical activation process, which causes the reconstruction of catalyst featured by the oxidation of Ni and Mn. The Ni 2*p* spectrum of the Ni₁Mn₁-MOF-Se/NF catalyst after activation indicates the disappearance of Ni⁰ and Mn²⁺ and the increase of Ni³⁺ and Mn⁴⁺ (Supplementary Figs. 9-11). The ICP-OES results of the electrolyte during the activation process also confirmed the dissolution of a small amount of Mn and Se into the electrolyte (Supplementary Table. 1).

Please check the supplemented discussions about the accumulation of Mn and Se elements after electrochemical activation on pages 4 and 5 in the revised manuscript.

Comment 2: The authors report in the SI (Figure S8) the ICP-OES results of the Ni₁Mn₁-MOF-Se/NF catalyst before and after activation. It would be insightful to add the ICP of the solution after electrocatalysis tests to see if the catalysts experienced any leaching.

Reply:

Thank you for the valuable comment. We have taken your suggestion to supplement the ICP-OES results of the electrolyte after electrochemical activation process and electrocatalysis test as shown in Supplementary Table 1. It can be found that the electrochemical activation causes partial dissolution of Mn and Se elements into the electrolyte. Additionally, the catalyst experienced some leaching of Mn and Se during the initial test followed by maintaining stable, indicating that the catalyst can work stably during the electrolysis test.

According to your suggestion, we have supplemented the ICP-OES results (Supplementary Table 1) and the related discussions on page 5 of the revised manuscript. Please check

Comment 3: The following references can be added as relevant examples of H₂O₂ electrosynthesis with N-doped carbon nanostructures. Enhanced on-site hydrogen peroxide electrosynthesis by a selectively carboxylated N-doped graphene catalyst.

ChemCatChem **13**, 4372-4383 (2021). N-doped graphitized carbon nanohorns as a forefront electrocatalyst in highly selective O₂ reduction to H₂O₂. *Chem* **4**, 106-123 (2018)

Reply:

Thank you for the valuable suggestion. These references are very helpful for discussing the catalytic mechanism and structure-activity relationship of N-doped carbon materials in H₂O₂ electrosynthesis. We have added them in the revised manuscript (Ref. 50 and Ref. 51). Please check

To Reviewer #4:

Comment 1: The authors state that they couple H₂O₂ production with PET upcycling but H₂O₂ generation reaction is coupled with the ethylene glycol oxidation. While ethylene glycol can be produced from PET, it is misleading to state that H₂O₂ is coupled with PET recycling.

Reply:

Thank you for the insightful comment. We have taken your suggestion to revise the related description to avoid the misunderstanding. In the first version of this work, we used the description like “coupling H₂O₂ production with PET upcycling” because that the ethylene glycol was directly obtained from the depolymerization of PET (in Figs. 3 and 6), rather than using the purchased one. Besides, the description of PET upcycling could also include the advantage of producing p-phthalic acid, the other product of PET depolymerization which don't participate in the oxidation process. The concept regarding coupling PET upcycling were also reported by previous works (*Nat. Commun.* 2021, 12, 4679; *J. Am. Chem. Soc.* 2023, 145, 6144-6155).

Please check the revisions as highlighted on page 1, 3 and 7 in the revised manuscript.

Comment 2: References for all the cost evaluations need to be provided. References for values in Table 6 must be also provided. It is not clear if the cost of all the chemicals (e.g. formic acid) considered in the calculation.

Reply:

Thank you for the beneficial comment. In the first submission, we added the references for the economic evaluation in the main text, while forgetting to add them in the section of the economic evaluation in the Supporting Information. To make it easy for reviewer and readers to understand the cost evaluation process, we have taken your suggestion to also add these references on pages 2-5 in the revised Supporting Information.

We have also added the references for values in Supplementary Table 6 (corresponding to Supplementary Table 7 in the revised Supporting Information).

For the raw material of formic acid (740 USD/ton, *Nat. Catal.* 2021, 4, 943-951) that was used in the potassium formate (KDF) synthesis process, we have taken your suggestion to re-calculate its cost and update the results of economic evaluation in the revised manuscript (page 17) and Supporting Information (page 4). The calculation results indicate that the ORR||EOR electrosynthesis system has a high gross profit of up to 50.8×10^6 USD, corresponding to an apparent increase of 104.0% and 134.1% than the HER||EOR system (24.9×10^6 USD) and the conventional ORR||OER system (21.7×10^6 USD), highlighting the market competitiveness and economic potential of the ORR||PET electrosynthesis system.

Additionally, considering that the electrolyte containing H₂O₂ can be directly used for the downstream production of sodium peroxyborate (SPB) and dibenzoyl peroxide (BPO), which avoids the separation problem. The converting from H₂O₂ to SPB and BPO can achieve profit growth. To simplify the economic evaluation and conduct parallel comparisons, we exclude both the raw material cost (borax, benzoyl chloride) and the product values (SPB, BPO).

We have taken your suggestions to provide these discussions on page 4 in the revised Supporting Information. Please check

Comment 3: During the activation of the catalyst, all the Se is lost according to the supplementary Figure 5. The authors should discuss this point. The authors should also discuss what the activation process consists of.

Reply:

Thank you for the valuable suggestions. The EDS mapping images of Ni₁Mn₁-MOF-

Se/NF (Supplementary Fig. 5) indicate that the electrochemical activation process causes the loss of surface Se. The XRD pattern (Supplementary Fig. 3) indicates that the Ni₁Mn₁-MOF-Se/NF catalyst after activation still shows the presence of NiSe and Ni_{0.85}Se. These results suggest that the Se element on the surface of the Ni₁Mn₁-MOF-Se/NF catalyst would be replaced by O during the activation process, while the Se element in the bulk phase maintains stable, which was also observed in the previous works (*Nat. Commun.* 2022, 13, 2916; *Adv. Funct. Mater.* 2020, 31, 2008812).

The electrochemical activation process of Ni₁Mn₁-MOF-Se/NF was conducted via the cyclic voltammetry (CV) method from -0.2 to 0.5 V (vs Hg/HgO) at scan rates of 50mV/s for 100 cycles in an electrolyte containing 1 M KOH and 0.5 M ethylene glycol. Due to the reduction of Ni and Mn by NaBH₄ during the selenization process to prepare Ni₁Mn₁-MOF-Se/NF, the main purpose of the electrochemical activation is to oxidize Ni and Mn to high valences. The XPS spectra and in situ Raman spectra imply that the electrode surface undergoes reconstruction and the valence state of Ni and Mn were changed during the activation process.

We have taken your suggestions to add these discussions on page 4 and 10 in the revised manuscript. Please check

Comment 4: EDX data should be provided with less significant digits.

Reply:

Thank you for the beneficial suggestion. We have revised the significant digits of EDX data in the revised Supporting Information. Please check

Comment 5: In supplementary figure 9a, an XPS peak is missing at about 878 eV. Why the intensity of the 2p_{1/2} peaks of Mn is so low, this has no physical sense. Mn²⁺ and Ni⁰ should provide doublets in the XPS spectra, but only the 2p_{3/2} peak is fitted.

Reply:

Thank you for the valuable comment. We have taken your suggestion to revise the Ni 2p XPS spectra in Supplementary Figs. 9a. According to the peak splitting criterion of XPS spectrum, the peaks from Ni 2p_{1/2} and Ni 2p_{3/2} should remain consistent. There

should be a peak at 878 eV that belongs to Ni⁴⁺, based on the peak splitting results in the first version of Supplementary Figs. 9a. By combining our experimental details with the literature, we confirm that the Ni-based catalysts involved in this work are mainly Ni²⁺ and Ni³⁺ (*Angew. Chem. Int. Ed.* 2021, 60, 10577-10582; *Nat. Energy* 2021, 6, 904-912). Therefore, the Ni 2*p* spectra in Supplementary Figs. 9a and Fig. 11a were re deconvoluted into six peaks, including Ni²⁺ (855.8 and 873.5 eV), Ni³⁺ (857.3 and 875.5 eV), and two satellite peaks.

The main reason for the low peak intensity of Mn 2*p*_{1/2} can be attributed to the XPS spectra deviation caused by the relatively low Mn content in the catalyst. Based on the characteristics of XPS, when the element content is less than 5%, the corresponding signal of the peak will be weak and can be influenced by interference from other elements. To solve this problem, we obtained the powder by ultrasonic from the foam nickel and then carried out XPS analysis again, and the accuracy of the results were improved (Supplementary Figs. 9b, 10b and 11b). We have also taken your suggestion to add both the 2*p*_{3/2} and 2*p*_{1/2} of Mn²⁺ and Ni⁰ in the XPS spectra according to the literature (*Nat. Catal.* 2022, 5, 109-118; *Adv. Energy Mater.* 2021, 11, 2003203).

We have taken your suggestions to revise the XPS spectra in Supplementary Figs. 9-11. Please check

Comment 6: Some references are misplaced (or do not refer to the text where they are inserted)

Reply:

Thank you for the valuable suggestion. We have taken your suggestions to revise the references as highlighted in the revised manuscript such as page 2. Please check

Comment 7: In figure 1a, H₂O should be H₂O₂

Reply:

Thank you for the beneficial suggestion. We have revised Fig. 1a in the revised manuscript. Please check

REVIEWER COMMENTS

Reviewer #1 (Remarks to the Author):

Yang, Qiu and co-workers have revised their contribution with respect to the original inaccuracies regarding the materials characterization. Unfortunately, I'm still not convinced about the authors' explanation with respect to the evident inconsistency between Figs. 4b and 4e in the electrochemical part. The difference in the current density values is huge (one order of magnitude) and it cannot be justified to iR compensation only. Authors should report the data in a coherent way before submitting their manuscript for evaluation, i.e. all electrochemical data should be iR compensated thus making any comparative data analysis coherent and consistent. In the present form, I regret that I cannot support this manuscript for publication in its current form.

Reviewer #2 (Remarks to the Author):

The comments have been addressed well, so I recommend publishing it now.

Reviewer #4 (Remarks to the Author):

The authors considered all the reviewers' suggestions and corrected the manuscript accordingly. My only remaining concern is the very low (0.0% according to supplementary figure 5) amount of Se measured after activation of the material according to SEM-EDS data. The authors state that the low Se amount measured is related to the surface oxidation of the material. However, EDS is not a surface-sensitive tool. The authors should reconsider this point or remeasure the Se content. Actually, the elemental maps show the presence of some Se. After reconsidering this point, the manuscript is ready for publication.

Response to Reviewers

Dear Reviewers:

Thanks a lot for your constructive comments and suggestions about our manuscript entitled “Energy-saving and product-oriented H₂O₂ electrosynthesis: from electrochemistry pairing to product conversion” (Manuscript number: NCOMMS-23-17553B). Those comments are all valuable and very helpful for improving the quality of our paper. We have studied your comments very carefully and substantially revised our manuscript in hope of addressing your concerns and meeting the high standards of *Nature Communications*. The comments were addressed point-by-point below and the involved changes have been highlighted in the revised manuscript and supporting information.

To Reviewer #1:

Comment 1: Yang, Qiu and co-workers have revised their contribution with respect to the original inaccuracies regarding the materials characterization. Unfortunately, I'm still not convinced about the authors' explanation with respect to the evident inconsistency between Figs. 4b and 4e in the electrochemical part. The difference in the current density values is huge (one order of magnitude) and it cannot be justified to *iR* compensation only. Authors should report the data in a coherent way before submitting their manuscript for evaluation, i.e. all electrochemical data should be *iR* compensated thus making any comparative data analysis coherent and consistent.

Reply:

Thank you for the valuable comments. We have taken your suggestion to supplement two confirmatory experiments, both of which match well with the data reported in Figs. 4b and 4e. On the one hand, experiments about the polarization curves of different catalysts for ORR-to-H₂O₂ in Fig. 4b were supplemented without *iR* compensation (Supplementary Fig. 28 in the revised Supporting Information), which match well with the current-time curves in Fig. 4e. On the other hand, we have also supplemented the current-time curves of B/N-onion carbon for ORR-to-H₂O₂ under different potentials in Fig. 4e under the condition of *iR* compensation (compensating resistance: $3.1 \pm 0.2 \Omega$,

Supplementary Fig. 30), which match well with the polarization curves in Fig. 4b with consideration the cliffy slope of the LSV curves. Moreover, other than the as-suggested data, we have also supplemented the potential-time curve at a current density of 100 mA cm⁻², which confirms the outstanding robustness of B/N-onion carbon for ORR under the condition of 100% *iR* compensation (compensating resistance: 3.2 ± 0.2 Ω, Supplementary Fig. 31).

According to the literature and the experimental results in this work, the difference in current density is related to the solution resistance between the working electrode and the reference electrode, which can be compensated by the *iR* compensation to some degree. The compensation levels reported in the literature usually range from 50% to 100%, and the impact of different compensation levels on the polarization curve also varies greatly. We selected an *iR* compensation level of 100% based on the literature about H₂O₂ electrosynthesis (*Nat. Commun.* 2021, 12, 4225; *Angew. Chem. Int. Ed.* 2022, e202206544; *Nat. Energy.* 2021, 6, 904-912). According to the valuable suggestion, we have also provided the specific compensation resistance values with error bar in all figure caption related to *iR* compensation for readers to derive E-*iR*.

We have taken your suggestions to supplement all the data and the corresponding discussions about the catalysis performance of B/N-onion carbon with and without *iR* compensation in the revised Supporting Information (Supplementary Figs. 13, 28, 30 and 31) and the revised manuscript (page 8, 9, 12, 13, 15, and 22). All original data have been added in the Source Data file. Please check

To Reviewer #4:

Comment 1: The authors considered all the reviewers' suggestions and corrected the manuscript accordingly. My only remaining concern is the very low (0.0% according to supplementary figure 5) amount of Se measured after activation of the material according to SEM-EDS data. The authors state that the low Se amount measured is related to the surface oxidation of the material. However, EDS is not a surface-sensitive tool. The authors should reconsider this point or remeasure the Se content. Actually, the elemental maps show the presence of some Se. After reconsidering this point, the manuscript is

ready for publication.

Reply:

Thank you for the insightful comment. We have taken your suggestions to supplement experiments about detecting the Se content using the X-ray photoelectron spectroscopy (XPS, mainly for surface region) and the inductively coupled plasma-optical emission spectrometer (ICP-OES, for both bulk and surface regions).

The supplemented XPS data have been provided (Supplementary Table 3), which shows that after the electrochemical activation, the atomic percentage content of surface Se element decreases from 17.88 at% to 7.79% at%, while the oxygen content increases from 21.34 at% to 37.22 at%.

The ICP-OES tests of Se content in the Ni₁Mn₁-MOF-Se powder before and after electrochemical activation have also been provided (Supplementary Table 1), which shows that the Se content maintains a high value despite the decreases from 56.82 wt% to 38.98 wt%.

Through combining the XPS analysis for surface region and the ICP-OES test for bulk and surface regions, it can be confirmed that the electrochemical activation process leads to the substitution of Se by O at the surface region, while the Se element in the bulk phase maintains stable, which is in a good accordance with the literature (*Nat. Commun.* 2022, 13, 2916; *Adv. Funct. Mater.* 2020, 31, 2008812).

We have taken your suggestions to provide these discussions about selenium content on page 5 in the revised manuscript. The ICP-OES and XPS results of the Ni₁Mn₁-MOF-Se/NF catalyst before and after activation are provided in the Supporting Information (Supplementary Fig. 5, Supplementary Tables 1 and 3). Please check